# DUSP4 protects BRAF- and NRAS-mutant melanoma from oncogene overdose through modulation of MITF

Nuria Gutierrez-Prat[1], Hedwig L Zuberer[1], Luca Mangano[1], Zahra Karimaddini[2], Luise Wolf[2], Stefka Tyanova[2], Lisa C Wellinger[4], Daniel Marbach[3], Vera Griesser[3], Piergiorgio Pettazzoni[1], James R Bischoff[1], Daniel Rohle[4,*], Chiara Palladino[1,*], Igor Vivanco[5,*]

MAPK inhibitors (MAPKi) remain an important component of the standard of care for metastatic melanoma. However, acquired resistance to these drugs limits their therapeutic benefit. Tumor cells can become refractory to MAPKi by reactivation of ERK. When this happens, tumors often become sensitive to drug withdrawal. This drug addiction phenotype results from the hyperactivation of the oncogenic pathway, a phenomenon commonly referred to as oncogene overdose. Several feedback mechanisms are involved in regulating ERK signaling. However, the genes that serve as gatekeepers of oncogene overdose in mutant melanoma remain unknown. Here, we demonstrate that depletion of the ERK phosphatase, DUSP4, leads to toxic levels of MAPK activation in both drug-naive and drug-resistant mutant melanoma cells. Importantly, ERK hyperactivation is associated with down-regulation of lineage-defining genes including *MITF*. Our results offer an alternative therapeutic strategy to treat mutant melanoma patients with acquired MAPKi resistance and those unable to tolerate MAPKi.

## Introduction

Melanoma progression and maintenance often depend on the continued stimulation of the RAS-RAF-MAPK signaling pathway. Genetic alterations of either NRAS (15–20%) or BRAF (40–50%) arise early during melanoma pathogenesis and are preserved throughout tumor progression (1). These mutagenic events render the two enzymes constitutively active, leading to unrestrained phosphorylation of downstream targets such as MEK and ERK. Although the therapeutic value of targeting mutant NRAS has yet to be clinically proven, several inhibitors targeting BRAF and MEK kinases have demonstrated rapid antitumor responses in most of the patients with activating BRAF mutations (2, 3). However, cells almost invariably adapt and develop resistance to these agents (4, 5). The switch from a drug-sensitive to a drug-resistant phenotype has been mainly attributed to the reactivation of the MAPK pathway, resulting in sustained drug-insensitive ERK signaling. Many mechanisms of acquired MAPK inhibitor (MAPKi) resistance have been described, including secondary mutations in *NRAS* or *MEK* genes, *BRAF* gene amplification, and *BRAF* alternative splicing (6). In addition, melanoma cells can acquire drug resistance by transitioning from an epithelial to a mesenchymal phenotype. The most common molecular changes associated with this transition are the suppression of microphthalmia-associated transcription factor (MITF) and up-regulation of receptor tyrosine kinases (RTKs), among others (7, 8). Some melanomas with acquired resistance to MAPKi become "addicted" to these drugs for their continued proliferation. In these tumors, chronic drug exposure results in hyperactivation of the oncogenic MAPK pathway, allowing cancer cells to maintain a residual level of signaling despite the presence of the inhibitor. Consequently, acute drug withdrawal causes an overwhelming and toxic level of pathway activation that results in tumor regression (4, 9, 10). These observations are consistent with the notion that oncogene overdose can be as much of a liability to the cancer cell as oncogene inhibition (11).

The phenomenon of oncogene overdose has been most clearly demonstrated in MAPKi-addicted cells, which has led to the proposal of using drug holidays as a therapeutic strategy to overcome drug resistance driven by pathway "super-activation" (10). However, it is unclear whether modulation of the ERK activity threshold could also impair cell viability in treatment-naive cells, raising the possibility of using this approach as an alternative therapeutic option.

Excessive ERK activity has been previously correlated with cell cycle arrest or cell death (12). For example, in prostate cancer cells, constitutive activation of RAF or MEK is sufficient to induce $G_0/G_1$ arrest, and ectopic expression of self-activating ERK mutants leads to cell growth impairment and differentiation (13, 14). These data

[1]Roche Pharma Research and Early Development, Oncology Discovery, Roche Innovation Center Basel, Basel, Switzerland   [2]Roche Pharma Research and Early Development, Informatics, Roche Innovation Center Basel, Basel, Switzerland   [3]Roche Pharma Research and Early Development, Pharmaceutical Sciences, Roche Innovation Center Basel, Basel, Switzerland   [4]Ridgeline Discovery Basel, Basel, Switzerland   [5]Institute of Pharmaceutical Science, King's College London, London, UK

Correspondence: igor.vivanco@kcl.ac.uk; chiara.palladino@roche.com; dan@charmtx.com
*Daniel Rohle, Chiara Palladino, and Igor Vivanco contributed equally to this work.

are consistent with the idea that a defined threshold of RAF/MEK/ERK pathway activity exists in different cell types that determines the phenotypic outcome of the signal (i.e., proliferation versus growth arrest). To maintain the appropriate level of cellular ERK signaling, the BRAF/MEK/ERK pathway is controlled by various feedback mechanisms. On one hand, ERK attenuates the activity of the MAPK pathway either through direct phosphorylation of upstream components or through the activation of other inhibitory kinases such as the ribosomal protein S6 kinase 1 (RSK1) and RSK2 (15). On the other hand, ERK activation also leads to the expression of multiple proteins that inactivate the pathway. For instance, mRNAs encoding several members of dual-specificity phosphatases (DUSPs), protein tyrosine phosphatases, and sprouty RTK signaling antagonists (SPRY) are rapidly transcribed in response to ERK signaling. In particular, DUSPs and protein tyrosine phosphatases selectively dephosphorylate and inactivate ERK, whereas SPRY proteins reduce the pool of GTP-loaded RAS (16). Considering that in fully transformed melanoma cells, the BRAF/MEK/ERK pathway is constitutively activated for proper tumor growth, and these feedback mechanisms are likely to play a critical role in keeping MAPK activation below the viability threshold. However, the importance of these inhibitory proteins in BRAF-mutant melanoma cells is still poorly characterized.

Here, we examined the role of several MAPK feedback loop regulators in drug-naive and MAPKi-resistant melanoma cell growth and survival. We found that depletion of the ERK phosphatase DUSP4 induces oncogene overdose and loss of fitness in both drug-naive and drug-resistant BRAF-mutant melanoma cell lines. Moreover, we demonstrate that drug-naive NRAS-mutant melanoma cells are also sensitive to DUSP4 silencing, thus broadening the therapeutic potential of targeting this phosphatase. Interestingly, we show that in both BRAF- and NRAS-mutated backgrounds, the detrimental effect of DUSP4 depletion is specific to MITF-expressing cells.

Overall, our data show that MAPK overdose driven by DUSP4 inactivation can effectively and selectively kill mutant melanoma cells with melanocytic identity. This raises the possibility of using this and other oncogene overdose–inducing strategies to not only treat patients with acquired MAPKi resistance but also as an alternative therapeutic approach to treat melanoma patients who are unable to tolerate MAPK inhibitors.

# Results

### Loss of the MAPK phosphatase DUSP4 is deleterious to BRAF$_{V600E}$ melanoma cells

Resistance to BRAF inhibitors in BRAF-mutant melanoma cells can occur through diverse molecular mechanisms that converge on reactivation of the BRAF-MEK-ERK signaling pathway. Interestingly, acute drug withdrawal leads to cell death–inducing hyperactivation of the oncogenic pathway (9). We therefore wanted to investigate whether super-activation of the MAPK pathway in treatment-naive BRAF-mutant cells was sufficient to induce cell death. To do this, we performed a focused siRNA-based screen in BRAF$_{V600E}$ melanoma

cells targeting either direct or indirect negative regulators of the BRAF-MEK-ERK pathway, where the effects of knockdown on cell growth were assessed over time. Although silencing of most genes had a mild growth inhibitory effect, down-regulation of DUSP4, a MAPK phosphatase, caused up to 43% cell growth impairment (Figs 1A and S1). We further validated the results from our screen by individually silencing the expression of DUSP4 and two other DUSP family members with roughly similar substrate selectivity (DUSP6 and DUSP10) in two different BRAF$_{V600E}$ melanoma cell lines. Interestingly, the growth inhibitory effects of DUSP4 knockdown were observed despite the compensatory increase in DUSP6 levels (Figs 1B and S2A and B). We extended these observations by analyzing the effects of DUSP4 knockdown in two additional BRAF$_{V600E}$ melanoma cell lines and found similar results (Figs 1C and S2C and D). To help rule out potential off-target effects, we established an additional cell model based on two different doxycycline-inducible DUSP4 shRNAs. Single-cell clones were isolated, and DUSP4 expression and cell growth were analyzed with or without doxycycline treatment. Again, the same cell growth impairment was observed in clones with strong DUSP4 down-regulation (Fig S2E and F). Of note, available data from publicly available genome-wide CRISPR-Cas9 genetic screens also confirmed that several human BRAF-mutant melanoma cell lines (https://depmap.org/ceres/) (17, 18) are sensitive to DUSP4 depletion.

To fully characterize the growth suppression phenotype observed upon DUSP4 down-regulation, we performed additional cell death and cell proliferation assays. We found that DUSP4 knockdown in four different BRAF-mutant melanoma cell lines induced significant proliferation arrest and apoptotic cell death as judged by dye dilution and Annexin V staining, respectively (Fig 1D and E). Overall, these results indicate that MAPK pathway de-inhibition through DUSP4 inactivation can potently impair the growth of BRAF-mutant melanoma cells.

### DUSP4 down-regulation induces oncogenic overdose through ERK overactivation in mutant melanoma cells

DUSP4 primarily functions as a negative feedback regulator of MAPK signaling. It can dephosphorylate ERK, JNK, and p38 MAP kinases at both phosphothreonine and phosphotyrosine residues (19, 20). To assess the impact of DUSP4 depletion on the activation of various MAPKs in cells, we measured the phosphorylated form of ERK and the two stress-activated MAPKs JNK and p38, after DUSP4 down-regulation in BRAF-mutant melanoma cells. Interestingly, DUSP4 silencing led to the selective accumulation of the phospho-form of ERK in the two cell lines analyzed (Fig 2A), suggesting that in melanoma cells, DUSP4 activity may be primarily directed toward ERK.

It has been suggested that in some tumor cells, too much oncogenic signal could impair cell survival (11). To evaluate whether the observed increased levels of phospho-ERK were responsible for the effects of DUSP4 knockdown on cell viability, we normalized ERK activation pharmacologically using subtherapeutic doses of two distinct inhibitors of ERK's upstream activator MEK (trametinib and cobimetinib) and analyzed the cell growth profiles. As shown earlier, DUSP4–down-regulated cells showed higher phospho-ERK levels that correlated with impaired cell growth. Strikingly,

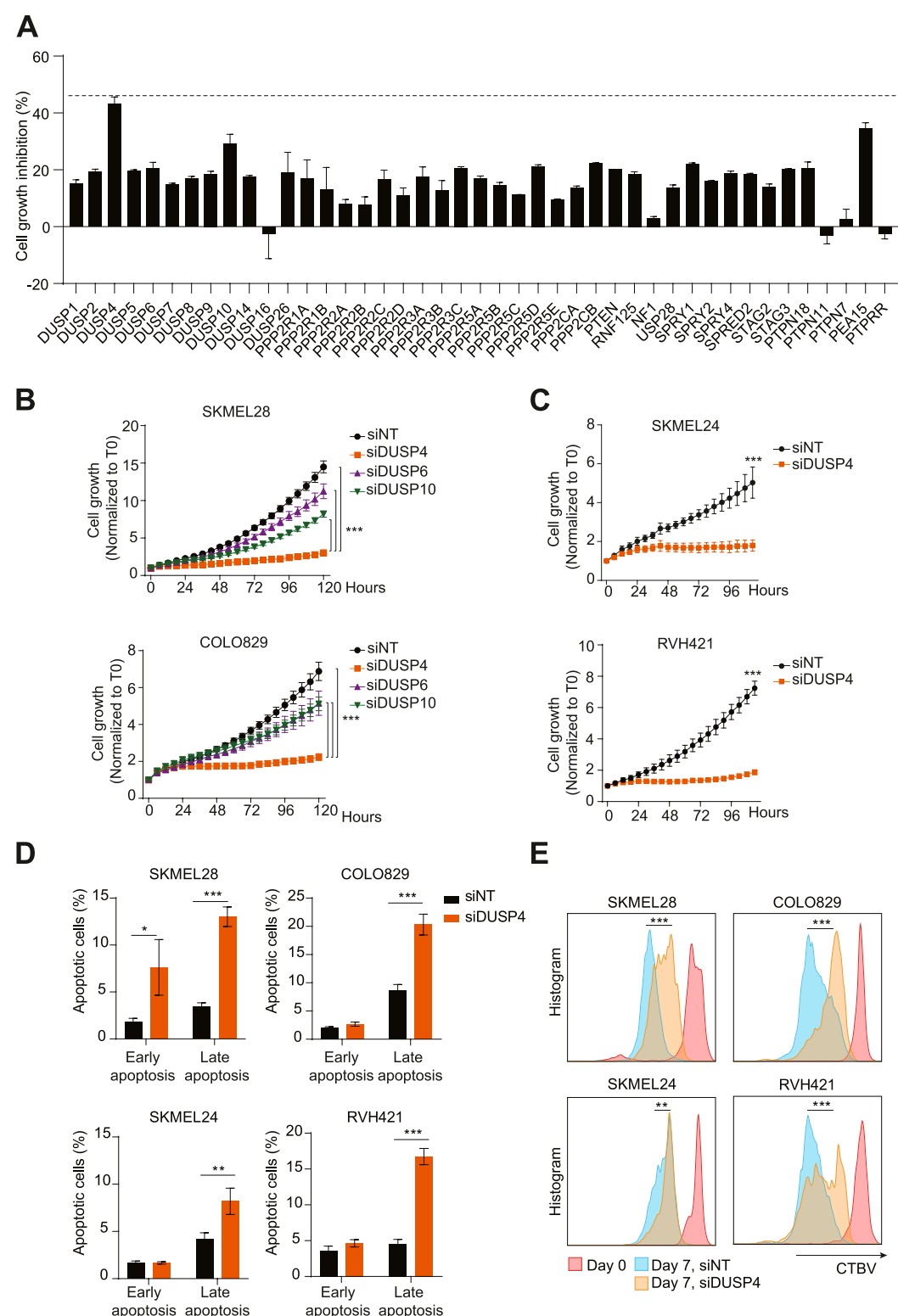

**Figure 1. DUSP4 deficiency leads to an impaired cell proliferation and cell death in mutant melanoma cells.**
**(A)** SKMEL28 cells were transfected with a customized siRNA library targeting the most known MAPK negative or crosstalk regulators as well as nontargeting control. The graph represents the percentage of cell growth inhibition normalized against nontargeting transfected cells, which were given the value of 0. The dashed line indicates the highest cell growth inhibition (up to 43%) observed upon DUSP4 down-regulation. Data represent mean ± SEM of three technical replicates. **(B, C)** SKMEL28, COLO829 (B), SKMEL24, and RVH421 (C) melanoma cells were transfected with indicated siRNAs against DUSP4, DUSP6, DUSP10, or with a nontargeting control. Graphs show the growth curves of transfected cells by measuring cell confluence over time. Cell growth values were normalized against time 0. Data are mean ± SEM, n = 3. **(B, C)** Statistical

normalization of phospho-ERK levels through MEK inhibitor treatment also reversed the growth inhibitory (Figs 2B–D and S3A–C) and apoptosis-inducing (Fig 2E) effects of DUSP4 knock-down, suggesting that ERK super-activation does indeed drive the observed antitumor effects. As expected, treatment with higher doses of MEK inhibitors led to a similar reduction in cell viability in both, control and DUSP4 knockdown cells (Figs 2B and C and 3A). Impor-tantly, changes in phospho-p38 levels were also detected in one of the two DUSP4–down-regulated cell lines (Fig 2A). However, normalization of p38 activity using a selective inhibitor (21) was not sufficient to restore cell growth, suggesting that p38 is unlikely to be involved in the observed growth inhibitory phenotype (Fig S3D and E). Taken together, these results indicate that in $BRAF_{V600E}$ melanoma cells, only a limited range of ERK activity is permissive for cell viability, such that potent inhibition or excessive activation can lead to a loss of fitness. The data also suggest that DUSP4 may be critical in maintaining the appropriate level of ERK activity in BRAF-mutant melanoma cells.

### DUSP4-mediated ERK activity controls the expression of MITF and its target genes in a lineage dependent fashion

To characterize the molecular intermediates responsible for the deleterious effects of DUSP4 loss, we performed an RNA-seq analysis of control and DUSP4–down-regulated cells. As our pre-vious experiments indicate that sublethal doses of MEK inhibitor can rescue the P-ERK levels and the effects of DUSP4 knockdown, we also included cells treated with a normalizing dose of trametinib in the analysis and compared gene expression profiles among all conditions. Interestingly, the expression of MITF (a master regulator of the melanocyte lineage) and some of its target genes were sig-nificantly down-regulated in the absence of DUSP4 and completely rescued by trametinib treatment (Fig 3A and Table S1). The *MITF* gene is expressed in different isoforms in which MITF-M (here referred as MITF) is exclusively expressed in melanocyte/melanoma cells and acts as a transcription factor controlling melanocyte development, survival, and differentiation (22). Given the importance of the *MITF* gene in the melanocytic lineage, we examined the MITF pathway in two $BRAF_{V600E}$ melanoma cell lines treated with the same conditions as in the RNA-seq experiment. Consistent with the transcriptome data, the mRNA levels of MITF and its target genes were reduced upon DUSP4 knockdown and rescued after MEK inhibition in the two cell lines analyzed by qPCR (Figs 3B and S4A).

It has been previously reported that RAF and MEK inhibitors can increase the expression of MITF and MITF targets in human mel-anoma cell lines (24, 25, 26). In fact, our qPCR data are consistent with these observations. This would suggest that the increased ERK activation upon DUSP4 knockdown might suppress MITF function. Of note, no changes in P-ERK levels and MITF expression were found upon knockdown of the related MAPK phosphatases DUSP6 and DUSP10 (Fig S4B and C). These results indicate that the expression

of MITF and its related target genes is mainly regulated by the DUSP4-ERK axis and might explain the deleterious effect upon DUSP4 deficiency in melanoma cells.

MITF function can be regulated by several transcription factors depending on cellular context and how ERK modulates *MITF* gene expression remains unclear (27). To better understand the role of the DUSP4-ERK axis in modulating MITF expression, we analyzed our RNA-seq data using the Virtual Inference of Protein-activity by Enriched Regulon (VIPER) algorithm. This computational tool provides an ac-curate assessment of the activity (rather than expression levels) of transcription factors by measuring the expression of their direct targets (28). In DUSP4–down-regulated cells, the transcription factors showing a significant increased activity were direct or indirect sub-strates of ERK such as ELK1 and CEBPB (29) (Fig 3C and Table S2), confirming that the higher phospho-ERK levels observed upon the loss of DUSP4 are indeed functional. In contrast, transcription factors found through VIPER analysis to have significantly lower activity after DUSP4 knockdown were related to the MITF pathway. Among these was PAX3, a transcription factor that has previously been described as a major upstream regulator of MITF (30). Interestingly, PAX3 activity was fully rescued by trametinib treatment (Fig 3C), suggesting that the DUSP4-ERK axis modulates the MITF pathway by suppressing PAX3 activity.

MITF has been previously implicated in melanoma survival pathways (31), suggesting that the suppression of *MITF* gene ex-pression could explain the defective cell growth observed in DUSP4 knockdown cells. To test this hypothesis, we silenced MITF ex-pression in mutant melanoma cells and evaluated cell growth over time. We found that MITF silencing largely phenocopied the cyto-static effects (and to a much lesser degree, the cytotoxic effects) of DUSP4 knockdown (Figs 3D and E and S5), suggesting that indeed the DUSP4 depletion–induced loss of MITF may be responsible for the growth inhibitory phenotype. As the expression of MITF is limited to the melanocytic lineage, we investigated whether the loss of DUSP4 could induce cell growth defects in other $BRAF_{V600E}$ cell lines derived from different tissues. Although all the cell lines tested were sensitive to BRAF knockdown, growth inhibition after DUSP4 silencing was only observed in melanoma cells despite a similar level of DUSP4 knockdown (Figs 3F and S4D), suggesting that the role of DUSP4 in supporting the viability of BRAF-mutant cells may be exclusive to the melanocytic lineage.

Taken together, these results indicate that DUSP4 selectively controls cell viability in BRAF-mutant melanoma cells by func-tioning as a rheostat for ERK signaling which in turn drives the PAX3-MITF pathway.

### The essential role of DUSP4 is restricted to MITF-expressing melanoma, and it is independent of the oncodriver mutation

Bioinformatic analyses of melanoma gene expression datasets revealed different subsets of BRAF/NRAS melanomas that differ in

---

significance was calculated between siDUSP4 cells and the other conditions either at 120 (B) or 102 (C) h. **(D)** Transfected cells were collected after 7 d and were stained with Annexin V and Zombie dyes to analyze cell death by FACS. Early apoptosis displays the percentage of Annexin V$^+$/Zombie$^-$ cells, whereas late apoptosis shows the percentage of Annexin V$^+$/Zombie$^+$ cells. Graphs show the quantification of three independent experiments. Statistical significance was calculated between the siNT and siDUSP4 conditions. **(E)** Transfected cells were stained with Cell Tracer Brilliant Violet (CTBV) and analyzed by CTBV incorporation after 15 min (day 0) or after 7 d (day 7). Different times and conditions are shown by different colors. Data are mean ± SEM, n = 3. Statistical significance was calculated between siNT and siDUSP4 cells at day 7.

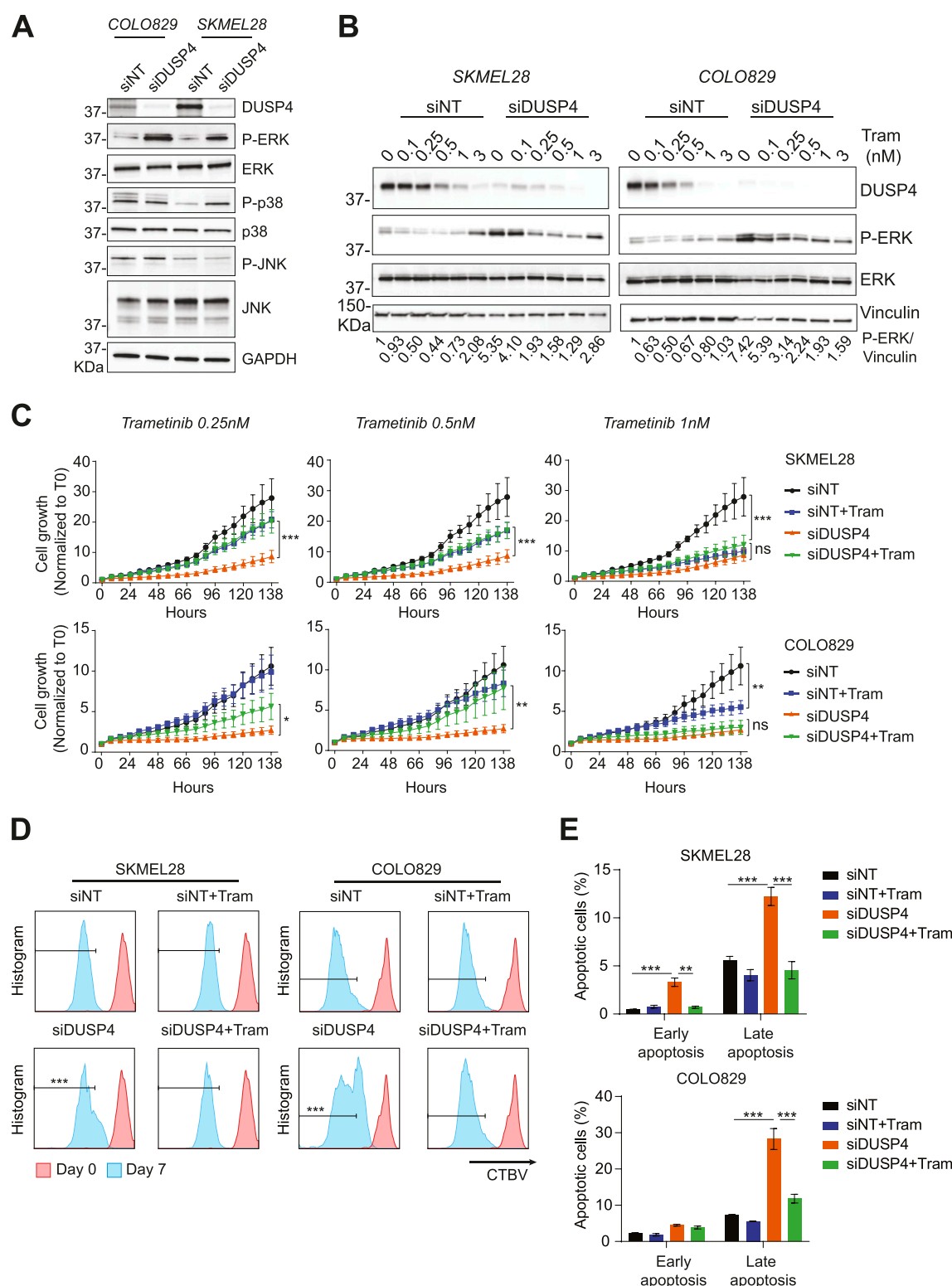

**Figure 2. ERK activation state modulates cell vulnerability upon DUSP4 knockdown.**
**(A)** Mutant melanoma cells (COLO829 and SKMEL28) were transfected with siRNA against DUSP4 or nontargeting control. 48 h later, cell lysates were analyzed by Western blot. **(B)** Transfected cells were treated with trametinib (Tram) at the indicated concentrations or with DMSO as a vehicle (0 nM). Lysates of the two melanoma cells were analyzed by immunoblot after 48 h of treatment. Band intensities were analyzed by ImageJ software, and the P-ERK/vinculin ratio is indicated. **(C)** Cells were treated as in (B), and the cell growth was analyzed by measuring cell confluence over time. Cell growth values were normalized against time 0. Data are mean ± SEM, n = 3. In the first two conditions (Tram 0.25 nM and Tram 0.5 nM), statistical significance was calculated between siDUSP4 versus siDUSP4+ Tram at 138 h. Samples treated with Tram 1 nM;

the endogenous expression of MITF independently of the onco-driver mutation. MITF-low melanomas were described to be more invasive and MITF-high melanomas more proliferative (32). As our data show that DUSP4 depletion leads to cell death through MITF suppression, we evaluated the sensitivity of BRAF/NRAS–mutant melanoma cells expressing either high or low levels of MITF to DUSP4 knockdown. As expected, DUSP4 silencing led to enhanced ERK phosphorylation in all cell lines tested (Fig 4A). However, ERK overactivation only correlated with reduced MITF levels and cell growth defects in melanoma cells with high basal levels of MITF (Figs 4A and B and S4E). Consistently, MITF silencing showed similar cell growth inhibition in high-MITF–expressing melanoma cells but no effect in low-MITF–expressing cells (Fig 4B), indicating that DUSP4 function is critical for both NRAS- and BRAF-mutant melanoma cells that rely on the MITF pathway. To further investigate whether the toxic effects of DUSP4 inhibition are dependent on MITF suppression rather than reaching a set threshold of ERK signaling alone, we raised the DUSP4-depleted levels of phospho-ERK even higher by concurrent knockdown of MAPK related phosphatases (DUSP6 and DUSP10) (Fig S6A and B). A further overactivation of ERK after DUSP4/DUSP6 (but not DUSP4/DUSP10) double knockdown was observed (Fig S6A). However, cell growth was only impaired in high-MITF–expressing cells in response to either dual DUSP4+6/10 or single DUSP4 inactivation (Fig S6C). These results are in line with DUSP4 depletion–induced ERK overactivation causing growth impairment through down modulation of the essential MITF pathway in MITF-proficient melanoma cells.

To confirm our findings in additional and more clinically relevant models, we extended these observations to low-passage patient-derived melanoma cells carrying NRAS mutations that have been profiled using mRNA, qPCR, and DNA short tandem repeat analysis to confirm that they closely match the original patient tumor tissue (Table S3). Consistent with our data from established cell lines, only cells with high MITF expression were sensitive to DUSP4 knockdown (Fig 4C and D), suggesting that the antitumor effects of dysregulating the DUSP4-ERK axis are specifically relevant to differentiated melanoma cells expressing high levels of MITF and with intrinsic mutations in the MAPK signaling pathway. Notably, in all human melanoma studies reported in cBioPortal, there is a significant positive correlation between DUSP4 and MITF mRNA expression (Fig 4E). Thus, in high-MITF tumors, DUSP4 is highly expressed likely because of its essential role in maintaining cell viability through restriction of MITF-inhibiting ERK activity. In contrast, in low-MITF tumors, where cell viability is less dependent of MITF function, DUSP4 activity is equally less relevant to cell viability, and consequently, these tumors tend to have lower levels of DUSP4 expression. This correlation from a large human melanoma dataset supports our hypothesis that MITF and DUSP4 are functionally co-

dependent. The existence of a distinct MITF expression profile opens the possibility of using it as a potential biomarker to stratify patients who are most likely to respond to anti-DUSP4 therapy.

## MITF expression levels in treatment-naive cells determine the essentiality of DUSP4 in MAPKi-resistant cells

The efficacy of therapies targeting the MAPK pathway in mutant melanoma is limited because of the development of resistance mechanisms that result in tumor relapse (34, 35). Although alternative treatments such as immunotherapy can benefit some patients, therapeutic options are limited for many others. Therefore, novel therapeutic interventions are still urgently needed for melanoma patients with acquired resistance to MAPKi (36). Using high-MITF (SKMEL28) or low-MITF (A375) drug-naive $BRAF_{V600E}$ melanoma cell lines, we derived MAPKi-resistant cell models to evaluate the role of DUSP4 in this setting. A trametinib-resistant SKMEL28 cell line (R+T cells) was generated by chronic treatment with lethal increasing concentrations of trametinib. A similar strategy was used to derive A375 cells resistant to vemurafenib as it was previously reported (37) (R+V cells). Importantly, our trametinib-resistant and vemurafenib-resistant cell lines exhibited cross-resistance to other MAPK inhibitors (Fig S7A and B) and to the combination of BRAFi and MEKi (Fig S7C and D). Although resistance was generated through chronic exposure to single-agent MAPK inhibitors, the fact that these cells are also resistant to the combination is consistent with the overlapping resistance mechanisms observed clinically in patients treated with single-agent or combination MAPKi therapy (38, 39). Interestingly, trametinib-resistant SKMEL28 cells were growth impaired when trametinib was acutely withdrawn (R–T acute) likely because of an enhanced P-ERK activation (4, 9) (Fig 5A and B). However, persistent drug discontinuation allowed these cells to regrow, whereas maintaining their resistance to all MAPKi tested (R–T sustain cells) (Figs 5A and S7A). In contrast, resistant cells derived from A375 (R–V cells) did not show the same drug addiction phenotype possibly because the ERK pathway was not sufficiently hyperactivated (Fig 5A and B).

We investigated the potential mechanisms of drug resistance in SKMEL28R + T and A375R + V cells and found no NRAS or MEK mutations (Fig S7E and F). However, SKMEL28 resistant cells showed a clear phenotype switch through the up-regulation of some RTKs such as AXL and EGFR as well as MITF silencing (Fig 5B). In contrast, A375 resistant cells showed BRAF amplification and a KRAS mutation (K117N) (37).

Importantly, DUSP4 levels in response to various conditions were also variable between the two cell lines tested. Although in A375 resistant cells DUSP4 was stably expressed among all conditions, in SKMEL28 resistant cells, DUSP4 levels were significantly down-regulated (R+T cells) initially but returned to basal upon

the significance was calculated between siDUSP4 versus siDUSP4+ Tram or siNT versus siNT+Tram at 138 h. **(D)** Transfected cells treated with or without Tram (0.25 nM) were stained with Cell Tracer Brilliant Violet (CTBV) and analyzed by CTBV incorporation the same day of the staining (day 0) or after 7 d (day 7). Different times are shown by different colors. Data are mean ± SEM, n = 3. Statistical significance was calculated between siDUSP4 versus siDUSP4+ Tram at day 7. **(E)** Transfected cells were treated with or without Tram (0.25 nM), and cell death was assayed by Annexin V and Zombie stainings. Early apoptosis indicates the percentage of Annexin V$^+$/Zombie$^-$ cells, whereas late apoptosis shows the percentage of Annexin V$^+$/Zombie$^+$ cells. Data are mean ± SEM, n = 3. Statistical significance was calculated between siDUSP4 versus siNT or siDUSP4+ Tram.

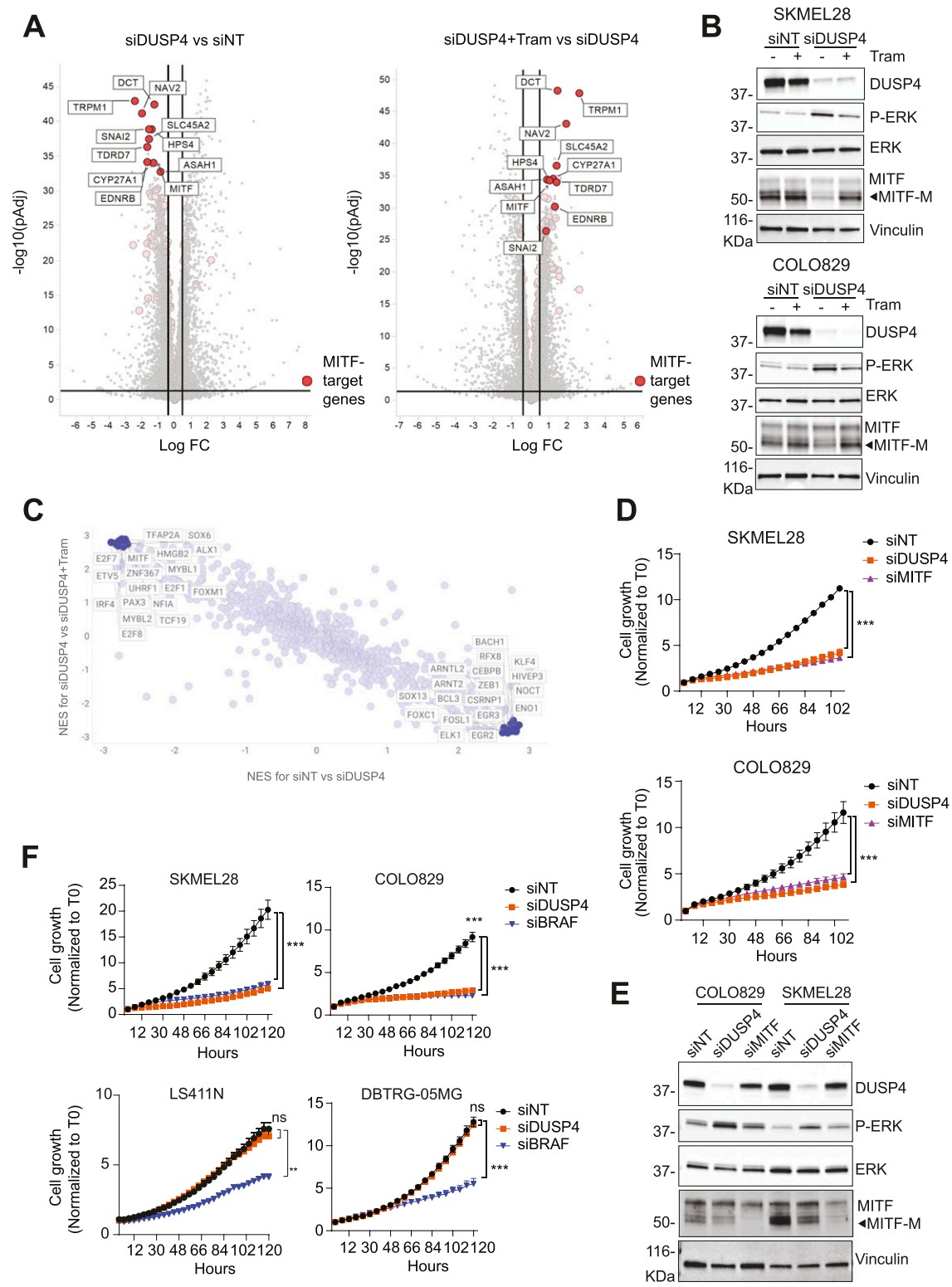

**Figure 3. DUSP4-dependent ERK activation leads to the suppression of the MITF pathway in mutant melanoma.**
**(A)** SKMEL28 cells were transfected with siRNA against DUSP4 or nontargeting control with or without trametinib (Tram) at 0.25 nM. After 48 h, cells were collected, and RNA-seq experiments were performed. Volcano plots display differentially expressed genes from the RNA-seq analysis between siDUSP4 and siNT or between siDUSP4+Tram and siDUSP4 (right panel). The vertical axis (y-axis) corresponds to the mean expression value of $\log_{10}$ (q-value), and the horizontal axis (x-axis) displays the $\log_2$ fold change value. Positive x-values represent up-regulated genes, and negative x-values represent down-regulated genes. MITF target genes are labeled in red (23). **(B)** Cells were treated as in (A), and 48 h later, lysates were analyzed by immunoblot. **(C)** Transcription factor activities were inferred based on the RNA-seq data using the

sustained drug withdrawal (R–T sustain cells) (Fig 5B). These observations led us to evaluate the effects of DUSP4 knockdown in MAPKi-resistant cell models (SKMEL28 R–T sustain, A375R + V and A375 R–V) compared with parental controls. Interestingly, resistant cells showed the same DUSP4 sensitivity profile as their parental counterparts. Thus, the loss of DUSP4 similarly impaired cell growth in SKMEL28 parental and resistant cells (Figs 5C–E), whereas no changes in cell growth were observed in any of the A375 cell models (Figs 5F–H). These results suggest that the role of DUSP4 is solely critical in high-MITF–derived cells with acquired resistance to MAPKi.

Of note, MITF expression was undetectable in trametinib-resistant SKMEL28 cells, possibly indicating a phenotypic switch which was also accompanied by the appearance of mesenchymal markers (N-CADHERIN, FRA1, and ZEB1) and an increase in AXL expression (Fig 5E). This is consistent with a previous report showing a strong association between a low MITF/AXL ratio and resistance to MAPK inhibitors (40). These data suggest that in these cells, DUSP4 may control cell viability through an MITF-independent mechanism (Fig 5B and E). Interestingly, in all our drug-resistant cells, DUSP4 inactivation triggered robust ERK hyperactivation similar to that seen in parental cells, suggesting that this phosphatase continues to function as a rheostat for MAPK kinase signaling in MAPKi-resistant cells (Fig 5E). To test this hypothesis, we compared the growth of DUSP4–down-regulated resistant and parental SKMEL28 cells treated with or without suboptimal doses of trametinib. Strikingly, similar to parental cells, trametinib treatment reversed DUSP4 knockdown–induced growth inhibition (and normalized phospho-ERK levels) in MAPKi-resistant cells (Fig 5I and J), indicating that vulnerability to DUSP4 inhibition in these cells is also dependent on the ability of DUSP4 to control ERK activity levels. Therefore, although resistant to MAPK inhibitors, these cells remain sensitive to inhibition of DUSP4 and the excessive ERK activation that follows.

# Discussion

The concept of oncogene overdose has been best characterized in drug addicted MAPKi-resistant melanoma cells (11). However, how oncogene overdose shapes the evolution of naïve/parental melanoma before treatment is still undetermined. Studies based on double-mutant engineered melanoma cells containing NRAS and BRAF activating mutations showed that the expression of the two genetic alterations can induce cell senescence because of an overactivation of the RAS/RAF/ERK pathway (41). This finding supports the idea that a "sweet spot" of oncogene activity defines a viability window in both MAPKi-resistant cells and their treatment-naive precursors.

Several factors are involved in maintaining the "sweet spot" of oncogenic signaling pathways, such as MAPK in melanoma. Although activating mutations in BRAF and NRAS amplify the output of the pathway, negative feedback signals impose limits on the level of this output (42). Therefore, it is tempting to hypothesize that altering the signaling output level in any direction could affect the homeostatic state of a cancer cell orchestrated by the driver oncogene. In this study, we have approached this hypothesis by silencing known negative regulators of the MAPK pathway and evaluating the effects of this perturbation on the survival of BRAF/NRAS-mutant melanoma cells. Among all genes analyzed, DUSP4 appeared to have a major role in controlling the viability of BRAF- and NRAS-mutant melanoma cells. Given that DUSP4 acts as a MAPK phosphatase, its role in the regulation of the BRAF-MEK-ERK signaling might be critical in controlling the magnitude of the signal. In fact, we found that the cell growth impairment linked to DUSP4 loss was rescued by restoring ERK signaling to its basal levels. Analogous observations have been reported in RTK-RAS-mutant lung adenocarcinoma, where DUSP6 inhibition causes cell toxicity through ERK overdose (43). All together, these observations highlight the importance of negative feedback regulation in maintaining viable levels of MAPK signaling in cells where this pathway is already mutationally activated. Interestingly, our results also identified PEA15 as a negative regulator of MAPK whose knockdown can inhibit cell growth, similar to DUSP4 inactivation (Figs 1A and S1). Because PEA15 is a cytoplasmic MAPK anchor that excludes activated ERK from the nucleus (44) and DUSP4 is a nuclear MAPK phosphatase, it is entirely possible that the nuclear fraction of hyperactivated ERK is the primary driver of oncogene overdose and the main cause of impaired cell viability in response to knockdown of either DUSP4 or PEA15. In this study, we decided to focus on DUSP4 as phosphatases represent a more drugable therapeutic target compared with scaffolding proteins. Our research reveals the molecular mechanism that links the hyperactivation of ERK signaling and loss of fitness after DUSP4 depletion in mutant melanoma cells. In particular, we found that the absence of DUSP4 in melanoma cells leads to excessive levels of active ERK. The aberrant activity of ERK is responsible for the suppression of the PAX3-MITF pathway, a key regulator of the melanocyte lineage, explaining the deleterious effects of DUSP4 loss. Furthermore, we show that knockdown of the highly related phosphatases DUSP6 and DUSP10 do not significantly affect cell viability or MITF levels (Fig S6). It is likely that the restricted nuclear localization of DUSP4 (as compared with DUSP6 and DUSP10) (45, 46, 47) and its preponderant expression levels (48) are important determinants of its specific role in restricting MAPK oncogene overdose. Given the lineage-

---

msviper (Virtual Inference of Protein-Activity by Enriched Regulon analysis) algorithm. The activity of transcription factors was analyzed comparing y-axis: NES for siDUSP4 versus siDUSP4+ Tram or x-axis: NES for siNT versus siDUSP4. Positive values in both axes represent increased activity, whereas negative values represent decreased activity. **(D)** Cells were transfected with siRNA against DUSP4, MITF, and nontargeting as the control. The cell growth was analyzed by measuring cell confluence over time. Cell growth values were normalized against time 0. Data are mean ± SEM, n = 3. Statistical significance was calculated between siNT versus siDUSP4 and siMITF conditions at 108 h. **(E)** Lysates from transfected cells were also analyzed by immunoblotting. **(F)** BRAF$_{V600E}$ cells from melanoma (SKMEL28 and COLO829), colorectal cancer (LS411N), and glioblastoma (DBTRG-05MG) were transfected with siRNA against BRAF and DUSP4. The cell growth was analyzed by measuring cell confluence over time. Cell growth values were normalized against time 0. Data are mean ± SEM, n = 3. Statistical significance was calculated between siNT versus siDUSP4 and siBRAF conditions at 120 h.

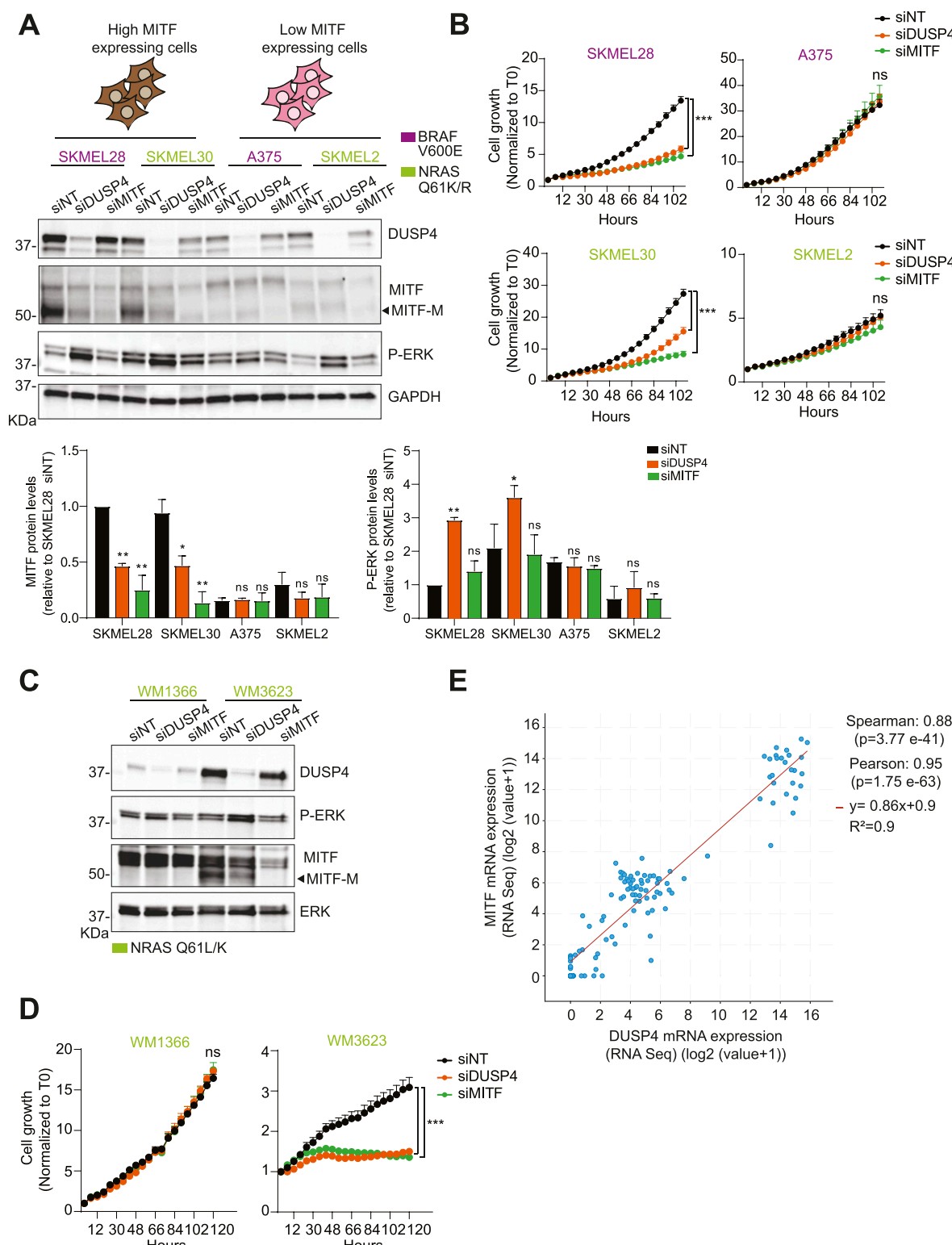

**Figure 4. Intracellular MITF levels determine the essential role of DUSP4 in melanoma independently of the oncodriver mutation.**
**(A)** BRAF$_{V600E}$ cells labeled in purple (SKMEL28, A375) and NRAS Q61K/R cells labeled in green (SKMEL30, SKMEL2, respectively) were transfected with siRNA against DUSP4, MITF, and nontargeting control. After 48 h, cell lysates were analyzed by Western blot. The upper cartoons classify melanoma cells according on their MITF expression levels. High levels of MITF are shown in brown, whereas low levels of MITF are labeled in pink. Band intensities were analyzed by ImageJ software, and the P-ERK/GAPDH and MITF/GAPDH ratios are indicated in the histogram. Data represent mean ± SEM of three independent experiments. Statistical significance was calculated against siNT for each different cell line. **(B)** Cells were treated as in (A), and cell growth was analyzed by measuring cell confluence over time. Graphs show cell

specificity of MITF, the role of DUSP4 in regulating cell viability is likely to be different in nonmelanocytic tumors. We assessed the response to DUSP4 depletion in two BRAF-mutant nonmelanoma cell lines (the glioma line DBTRG and the colorectal carcinoma cell line, LS411N) that were sensitive to MAPK inhibitors (Fig S8). Consistent with our prediction, we found that DUSP4 silencing did not affect their growth. Also consistent with this prediction, DUSP4 genomic loss has been associated with pancreatic tumor progression (49), and in some colorectal carcinomas, DUSP4 downregulation leads to cell proliferation and invasiveness (50).

In melanoma development and progression, the role of MITF is rather ambiguous. In some cases, high levels of MITF have been associated with terminal differentiation and cell cycle arrest (51). However, it has also been found that 15% of metastatic melanomas carry MITF gene amplification (52). Moreover, some melanoma cells do not express MITF and display invasive properties (53). In an attempt to reconcile these findings, a rheostat model has been proposed (54). Thus, three different phenotypes depend on the levels of MITF, ranging from differentiation (high MITF), proliferation (moderate MITF), and invasion (low MITF). The heterogeneity of MITF levels in melanoma restricts the essential role of DUSP4 to high- and moderate-MITF–expressing cells. Thus, independently of the activating lesion, melanoma cells with mutationally active MAPK and a functional MITF pathway are vulnerable to DUSP4 depletion.

Despite the effectiveness of current therapeutic options for melanoma patients (55), drug resistance and systemic toxicities limit the long-term efficacy of such treatments. The best characterized mechanism of resistance to MAPKi is the reactivation of the MAPK pathway by alterations in the MAPK pathway itself (39, 56, 57). In some cases, this reactivation sensitizes melanoma cells to drug discontinuation, which results in toxic levels of MAPK signaling. DUSP4 levels have been shown to be down-regulated in MAPKi-resistant cells (9). We find that DUSP4 levels are restored after drug withdrawal, suggesting that this phosphatase is part of a tightly regulated feedback control mechanism. In the presence of drug, BRAF-mutant melanoma cells appear to suppress DUSP4 expression to compensate for the acute loss of oncogenic signals. However, upon treatment discontinuation, DUSP4 levels need to be up-regulated once more to prevent toxic levels of pathway activation.

High-level focal amplification of MITF has been found to mediate BRAF inhibitor resistance in melanoma (39, 58). Paradoxically, the absence of endogenous MITF expression has also been be associated with MAPK inhibitor resistance (8), suggesting that either the excess or absence of MITF is permissive for cell viability under decreased MAPK signaling in BRAF-mutant melanoma. Our data show that both MITF-high and MITF-low BRAF-mutant melanoma cells can acquire MAPKi resistance (Fig S7A and B) and that their response to DUSP4 depletion remains unchanged (Fig 5C and F)

despite changes in MITF levels during the acquisition of drug resistance. For example, MAPKi-resistant cells derived from MITF-high SKML28 cells lose MITF expression becoming trametinib-resistant. Importantly, this phenomenon correlates with a phenotype switch that involves the acquisition of mesenchymal character and the up-regulation of RTKs such as AXL and EGFR. The latter has been shown by others to correlate with acquired drug resistance (8) and the mesenchymal switch likely explains the ability of these cells to survive in the absence of MITF. These cells display a drug-addicted phenotype and die upon acute trametinib removal. Surprisingly, we found that cells that survive after long-term drug withdrawal remained sensitive to DUSP4 depletion but only if the parental cell line from which they originated was MITF-high. We suspect that DUSP4 sensitivity in this case is likely associated with ERK-dependent downstream targets other than MITF or that the MITF-dependent factors that regulate sensitivity to super-activation of ERK are maintained by activated RTKs and/or driven by the mesenchymal phenotype. Considering that melanoma cells with de novo MITF deficiency were consistently insensitive to DUSP4 depletion regardless of MAPKi sensitivity status, assessment of MITF levels in treatment-naive melanoma tumors would likely be necessary to stratify patients who are most likely to respond to DUSP4-targeted therapies.

# Materials and Methods

### Cell lines

Most of the cell lines used were provided by the Roche Non-Clinical Biorepository (from F Hoffmann-La-Roche Basel). SKMEL28 and SKMEL2 were cultured in MEM Glutamax 41090 (# 41090028; Thermo Fisher Scientific) supplemented with 10% FBS (#97068-085; VWR). SKMEL24 were maintained in MEM Glutamax 41090 (# 41090028; Thermo Fisher Scientific) supplemented with 15% FBS (#97068-085; VWR), 1x NEAA (#11140050; Thermo Fisher Scientific) and sodium pyruvate. COLO829, SKMEL30, RVH421, DBTRG-05MG, and LS411N cells were cultured with RPMI 1640 (#A10491; Thermo Fisher Scientific) supplemented with 10% FBS (#97068-085; VWR). A375 were cultured with DMEM high glucose (#41966; Thermo Fisher Scientific) + 10% FBS (#97068-085; VWR). The SKMEL28-resistant cell line was obtained by chronically treating them with lethal increasing concentrations (0.5–20 nM) of trametinib until resistant pools showed up, whereas A375 cells resistant to 2.5 $\mu$M vemurafenib were obtained as previously described (37). Patient-derived melanoma cell lines (Table S3) were provided by the Wistar Institute and cultured with 80% of MCDB 153 media (#P04-80062; Pan Biotech), 18% Leibovitz's L-15 media (# 11415064; Thermo Fisher Scientific), 2% FBS, 1.68 mM CaCl$_2$, and 6 mM Glutamine. All cells used

---

growth curves normalized against time 0. Data are mean ± SEM, n = 3. Statistical significance was calculated between siNT versus siDUSP4 condition at 102 h. **(C)** Patient-derived cell lines containing NRAS Q61L/K mutation (WM1366 and WM3623, respectively) were transfected as in (A). After 48 h, cell lysates were analyzed by Western blot. **(D)** WM1366 and WM3623 cells were treated as in (A), and cell growth was analyzed by measuring cell confluence over time. Graphs show cell growth curves normalized against time 0. Data are mean ± SEM, n = 3. Statistical significance was calculated between siNT versus siDUSP4 condition at 120 h. **(E)** Co-expression analysis of DUSP4 and MITF in human melanoma. Bivariate and rank correlation analysis of combined gene expression data (n = 124) from four separate studies available through cBioportal (33).
Source data are available for this figure.

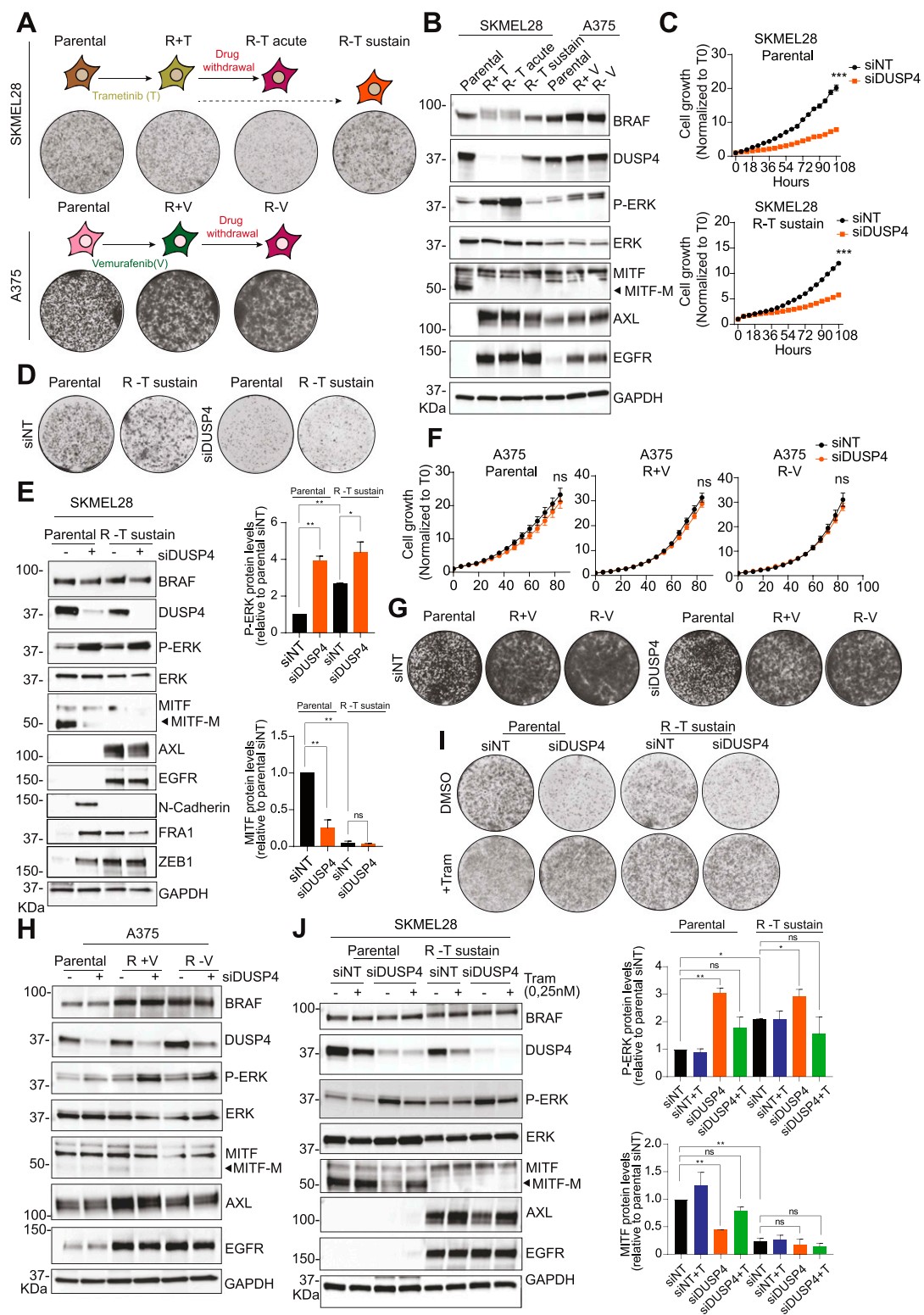

**Figure 5. The critical role of DUSP4 in MAPKi-resistant cells is strictly linked to the MITF expression levels in parental melanoma cells.**
**(A)** Schematic representation of the generation of SKMEL28- and A375-resistant cells. Parental SKMEL28 cells were treated with increasing doses of trametinib (up to 20 nM) until they acquired resistance (R+T cells). Trametinib was removed from R+T in an acute (R–T acute) or sustain (R–T sustain) manner. Parental A375 cells were treated with increasing doses of vemurafenib (up to 2.5 μM) until they acquired resistance (R+V cells). Vemurafenib was removed from R+V cells in an acute way (up to 6 d, R–V cells). Clonogenic growth of parental and resistant cells is shown. **(B)** Lysates of SKMEL28 and A375 parental and resistant cells were analyzed by immunoblot. **(C)** SKMEL28 parental and resistant cells with a sustain drug withdrawal (R–T sustain) were transfected with siRNA against DUSP4 and nontargeting as the control, and the

were incubated at 37°C, 90% humidity and 5% CO$_2$. For passaging, cells were washed in PBS and incubated in trypsin at 37°C until detached. Then, complete media was added, and cells were diluted as desired depending on the confluence and replated in a new culture dish.

### Cell treatments

For the inhibition of MEK, we used trametinib (# S2673; Selleckchem) and cobimetinib (#S8041; Selleckchem) at indicated concentrations. Vemurafenib (#S1267; Selleckchem) and dabrafenib (#2807; Selleckchem) inhibited BRAF at indicated concentrations. For the inhibition of the p38 pathway, we used Skepinone-L (#S7214; Selleckchem) at indicated concentrations. Activation of the p38 pathway was induced by 100 $\mu$M hydrogen peroxide (H$_2$O$_2$) (#H1009; Sigma-Aldrich) for 1 h.

### siRNA screening of MAPK negative regulators

A list of negative regulators of the BRAF/MEK/ERK pathway was placed into an siRNA mini-library using the Cherry-Pick Custom Library Tool (Dharmacon) (Table S4). A total of 50 nM siRNA was introduced into SKMEL28 cells by reverse transfection using Lipofectamine RNAiMAX Transfection Reagent (#13778150; Thermo Fisher Scientific) and Opti-MEM (#51985026; Thermo Fisher Scientific). A total of 10,000 cells/well were seeded in a 96-well plate in a final volume of 100 $\mu$l. Cell growth was analyzed with the Incucyte S3 Live-cell Analysis System (Essen BioScience) by acquiring four images per well for 3 d. Cell growth inhibition was calculated and normalized against the siNT control.

### Genetic depletion by siRNA

Depending on the experiment, 3 × 10$^5$ or 9 × 10$^5$ cells were seeded in a six-well plate or in a 10-cm dish, respectively. Reverse transfection using Lipofectamine RNAiMAX Transfection Reagent (#13778150; Thermo Fisher Scientific) and Opti-MEM (#51985026; Thermo Fisher Scientific) was performed according to the siTOOLS Biotech protocol using 3 nM final concentration of siPOOL nontargeting control and siPOOL against DUSP4, DUSP6, DUSP10, MITF, BRAF (siTOOLs Biotech). After 24 h, cells were collected and seeded at 2.5 × 10$^4$ cells/ml in a 96-well plate, and cell growth was assessed with the Incucyte S3 Live-cell Analysis System (Essen BioScience) by

acquiring four images per well for 7 d. After 48 h, the rest of the cells were analyzed by Western Blot or qPCR.

### Cell growth analysis

Cell growth was determined by the measure of cell confluence over time. Four pictures/well were taken by the Incucyte S3 Live-Cell Analysis System (Essen Bioscience) every 6 h, and data were analyzed using the Incucyte S3 Live-Cell Analysis System looking at the cell confluence per image.

### Cell viability assay

A total of 2.5 × 10$^4$ cells/ml were plated in a 96-well plate. After 24 h, cells were treated with a dose response of BRAF (10-0.00152 $\mu$M, 1:3 dilution), MEK inhibitors (1-0.000152 $\mu$M, 1:3 dilution), or the corresponding controls (0.15% DMSO). Cell viability was measured after 72 h using CellTiter-Glo 2.0 Cell Viability Assay (#G9242; Promega).

### Cell tracer-based proliferation assay

The CellTrace Violet Cell Proliferation Kit (CTBV, # C34571; Thermo Fisher Scientific) was used to measure cell proliferation. Each time that a cell divides, CTBV is transferred equally among the daughter cells, reducing fluorescence in half. Cells were labeled with CTBV following manufacture's protocol. Once all cells were stained, 1 × 10$^5$ cells were analyzed by flow cytometry (day 0), whereas the rest were transfected either with siDUSP4 or siNT and plated in a six-well plate for 7 d (day 7). Cells that reached 80% of confluence were transferred into a 10-cm dish to let them proliferate normally. Trametinib (0.25 nM) or DMSO was added immediately after transfection and after changing the media every 2–3 d.

### Annexin V and zombie stainings

Annexin V and Zombie staining ware used as a measurement of cell death. A total of 3 × 10$^5$ cells were harvested, washed once with PBS, and centrifuged. Pellets were resuspended with PBS, and 1 × 10$^5$ cells were plated in triplicates into a 96-well plate. First, cells were stained with 1:500 Zombie NIR (#423105; BioLegend) for 20 min at 4°C. Then, cells were washed (PBS + 0.5% BSA + 2 mM EDTA) and incubated with 1:100 Annexin V FITC (# V13242; Thermo Fisher Scientific) for 20 min at RT in the dark. Up 10,000 cells were acquired on

cell growth was analyzed by measuring cell confluence over time. Cell growth values were normalized against time 0. Data are mean ± SEM, n = 3. Statistical significance was calculated between siDUSP4 versus siNT condition at 108 h **(D)** SKMEL28 parental and resistant cells were treated as in C, and cell growth was analyzed by a colony formation assay. Representative images are shown from three independent experiments. **(E)** SKMEL28 parental and resistant cells were treated as in (C) and after 48 h, and cell lysates were analyzed by Western blot. Band intensities were analyzed by ImageJ software, and the P-ERK/GAPDH and MITF/GAPDH ratios are indicated in the histogram. Data represent mean ± SEM of three independent experiments. Statistical significance was calculated against siNT. **(F)** A375 parental and resistant cells were transfected with siRNA against DUSP4 and nontargeting as the control, and the cell growth was analyzed by measuring cell confluence over time. Cell growth values were normalized against time 0. Data are mean ± SEM, n = 3. Statistical significance was calculated between siDUSP4 versus siNT conditions at 84 h. **(F, G)** A375 parental and resistant cells were treated as in (F), and cell growth was analyzed by a colony formation assay. Representative images are shown from three independent experiments. **(H)** A375 parental and resistant cells were treated as in F, and after 48 h, cell lysates were analyzed by Western blot. **(I)** SKMEL28 parental and R–T sustain cells were transfected with siRNA against DUSP4 and nontargeting control in the presence or absence of trametinib (0.25 nM). Cells were plated to form colonies and analyzed 7 d later. Representative images are shown from three independent experiments. **(J)** SKMEL28 parental and R–T sustain cells were treated as in (I) and cell lysates were analyzed by Western blot after 48 h post-transfection. Band intensities were analyzed by ImageJ software, and the P-ERK/GAPDH and MITF/GAPDH ratios are indicated in the histograms. Data represent mean ± SEM of three independent experiments. Statistical significance was calculated against siNT.

a CytoFLEX S Benchtop Flow Cytometer, and FlowJo_V10 was used for analysis.

## Colony formation

A total of $1 \times 10^4$ of SKMEL28 and A375 cells were plated in triplicate in six-well plates and cultured for 6 d. Then cells were fixed in 4% paraformaldehyde (#11481745; MP Biomedicals) and stained with crystal violet (#HT90132-1L; Sigma-Aldrich).

## Western Blot

Cells were collected using Pierce IP Lysis Buffer (# 87788; Thermo Fisher Scientific) supplemented with Thermo Fisher Scientific Halt Protease Inhibitor Cocktail (# 78425; Thermo Fisher Scientific). Upon cell lysis, protein concentration was assessed with the DC Protein Assay (#5000112; Bio-Rad), normalized, and Laemmli Buffer (#J61337; AlfaAeser) added. Total protein lysates (30 µg) were separated on SDS–PAGE using 4–15% Criterion TGX Precast Midi Protein Gel (Bio-Rad) and transferred to a Trans-Blot Turbo Midi Nitrocellulose membrane (#1704159; Bio-Rad). Then, membranes were blocked using 1× of Animal-Free Blocking Solution (#15019; Cell Signaling) in TBS 0.01% Tween for 1 h at RT. After blocking, membranes were incubated at 4°C overnight with primary antibodies diluted with blocking buffer. The following antibodies were used: DUSP4 (# 5149, 1:1,000; Cell Signaling), DUSP6 (# 39441, 1:500; Cell Signaling), MITF (#12590, 1:500; Cell Signaling), Phospho-Erk1/2 (Thr202/Tyr204) (#4370, 1:1,000; Cell Signaling), Erk1/2 (#4695, 1:1,000; Cell Signaling), Phospho-JNK (Thr183/Tyr185) (#9251, 1:1,000; Cell Signaling), Phospho-HSP27 (Ser82) (#2401, 1:250; Cell Signaling), HSP27 (#95357, 1:1,000; Cell Signaling), Phospho-p38 MAPK (Thr180/Tyr182) (#4511, 1:1,000; Cell Signaling), p38 MAPK (#9212, 1:1,000; Cell Signaling), EGFR (#2232S, 1:1,000; Cell Signaling), BRAF (#14814S, 1:1,000; Cell Signaling), phospho-AKT (Ser473) (#9271; Cell Signaling), FRA1(#5281; Cell Signaling), ZEB1 (#3396; Cell Signaling), N-Cadherin (#13116; Cell Signaling), AXL (#8661S, 1:1,000; Cell Signaling), GAPDH (#5174, 1:2,000; Cell Signaling), and vinculin (#13901, 1:1,000; Cell Signaling). The Secondary anti-rabbit IgG, HRP-linked antibody (# 7074S; Cell Signaling) was diluted with blocking buffer at 1:5,000, and protein bands were visualized with the Western Bright Sirius HRP Substrate (Advansta #K-12043) using Fusion FX (Vilber Lourmat).

## qRT-PCR

Cell pellets were lysed with the RNeasy Mini Kit (#74104; QIAGEN) following manufacture's protocol. RNA levels were assessed using qScript XLT One-Step RT-qPCR ToughMix (#95132-100; Quanta Bioscience) and the following TaqMan Gene Expression Assays: DUSP4 (# Hs01027785_m1; Thermo Fisher Scientific), DUSP6 (# Hs04329643_s1; Thermo Fisher Scientific), DUSP10 (# Hs00200527_m1; Thermo Fisher Scientific), MITF (# Hs01117294_m1; Thermo Fisher Scientific), DCT (#Hs01098278_m1; Thermo Fisher Scientific), TYRP1(#Hs00167051; Thermo Fisher Scientific), TRPM1 (#Hs00931865_m1; Thermo Fisher Scientific), BCL2 (#04986394_s1; Thermo Fisher Scientific), EDNRB (#Hs00240747_m1; Thermo Fisher Scientific), and GAPDH (# Hs02786624_g1; Thermo Fisher Scientific). The qRT–PCR was performed in the LightCycler 480 System

(Roche). The $2^{\Delta\Delta Ct}$ method was used to calculate the relative RNA expression and normalized to GAPDH control.

## mRNA sequencing of SKMEL28 cells

3 nM siRNA against DUSP4 or nontargeting control were introduced in SKMEL28 cells by reverse transfection using Lipofectamine RNAiMAX Transfection Reagent (#13778150; Thermo Fisher Scientific) and Opti-MEM (#51985026; Thermo Fisher Scientific). A total of $3 \times 10^5$ cells were seeded in a six-well plate. In parallel, either 0.25 nM trametinib or vehicle (DMSO) was added in both siDUSP4- and siNT-transfected cells. After 16 h, cells were harvested and frozen at –80°C. RNA was extracted using the QIAGEN RNeasy Mini Kit (#74104), following manufacturer's instructions except that lysates were loaded on a QIAshredder column before loading on a spin column. Sequencing libraries were generated from 100 ng input RNA using the Illumina TruSeq Stranded mRNA LT Sample Preparation Kit (Set B, #RS-122-2102) as per manufacturer's instruction. Libraries were sequenced on an Illumina HiSeq4000 using paired end sequencing 2 × 50 bp reads to an average depth of 18 to 37 million sequences per sample. Base calling was performed with BCL to FASTQ file converter bcl2fastq:2.19 from Illumina. To estimate gene expression levels, paired-end RNASeq reads were mapped to the human genome (hg38) with STAR aligner (v2.5.2a) using default mapping parameters (59). Numbers of mapped reads for all RefSeq transcript variants of a gene (counts) were combined into a single value by using SAMTOOLS software (60). Differential expression analysis between the three replicates of cell lines from two different conditions were conducted using the Rsubread and the edgeR quasi-likelihood (QL)pipeline (61). Briefly, lowly expressed genes were first removed by only keeping genes which were expressed at 0.7 counts per million in three of six samples per comparison. The filtered count matrix was then sent through the edgeR QL to model the mean/variance trend of the read counts before differential expression analysis. The QL F-tests was used instead of the more usual likelihood ratio tests (LRT) as they give stricter error rate control by accounting for the uncertainty in dispersion estimation. To control the false discovery rate, multiple testing correction was performed using the Benjamini–Hochberg method. Hence, all statistics reported from the RNA-seq data in this manuscript are the false discovery rate adjusted *P*-values (Q values).

## Detection of MEK1 and NRAS mutations

For identification of MEK1 and NRAS mutations in SKMEL28- and A375-resistant cell lines, genomic DNA was isolated, and PCR was performed with primers amplifying *MEK1* (exon 2-Fw 5'-TGATGAG-CAGCAGCGAAAGC-3', exon 2-Rv 5'-GAACACCACACCGCCATTGC-3') and *NRAS* (exon 3 Fw 5'-TGGCAAATACACAGAGGAAGC-3', exon 3 Rv 5'-CACACCCCCAGGATTCTTAC-3'). Mutations were evaluated by Sanger Sequencing.

## Statistical analysis

Data are expressed as average ± SEM. Statistical analysis was performed by using *t* test for the comparison of two groups or

ANOVA for multiple groups using GraphPad Prism Software 7 (GraphPad Software, Inc.). *P*-values are expressed as \*$P \leq 0.05$, \*\*$P \leq 0.01$, and \*\*\*$P \leq 0.001$.

## Data Availability

The RNA-seq data from this publication have been deposited to the Gene Expression Omnibus database (https://www.ncbi.nlm.nih.gov/geo/) and assigned the identifier GSE181467.

## Supplementary Information

## Acknowledgments

The authors thank Roche Non-Clinical Biorepository for providing cell lines used for this work and the Roche Flow Cytometry Facility for the technical support and Dr. Jorge Vialard for the scientific discussion in the course of this work. N Gutierrez-Prat was supported by the Roche Postdoctoral Fellowship Program.

### Author Contributions

N Gutierrez-Prat: data curation, formal analysis, investigation, methodology, and writing—original draft, review, and editing.
HL Zuberer: investigation and methodology.
L Mangano: investigation and methodology.
Z Karimaddini: data curation, software, and formal analysis.
L Wolf: data curation, software, and formal analysis.
S Tyanova: data curation, software, and formal analysis.
LC Wellinger: investigation.
D Marbach: data curation, software, and formal analysis.
V Griesser: investigation.
P Pettazzoni: investigation and methodology.
JR Bischoff: resources and project administration.
D Rohle: supervision and funding acquisition.
C Palladino: supervision, funding acquisition, investigation, project administration, and writing—review and editing.
I Vivanco: supervision and writing—review and editing.

### Conflict of Interest Statement

N Gutierrez-Prat, HL Zuberer, L Mangano, Z Karimaddini, L Wolf, S Tyanova, D Marbach, V Griesser, P Pettazzoni, JR Bischoff, and C Palladino are current employees of F Hoffmann-La Roche Ltd. LC Wellinger and D Rohle are former employees of F Hoffmann-La Roche Ltd.

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
