## [Reviewer comments · Life Science Alliance]

Life Science Alliance

DUSP4 protects BRAF and NRAS-mutant melanoma from oncogene overdose through modulation of MITF

Nuria Gutierrez Prat, Hedwig Zuberer, Luca Mangano, Zahra Karimaddini, Luise Wolf, Stefka Tyanova, Lisa Wellinger, Daniel Marbach, Vera Griesser, Piergiorgio Pettazzoni, James Bischoff, Daniel Rohle, Chiara Palladino, and Igor Vivanco
DOI: <https://doi.org/10.26508/lsa.202101235>

Corresponding author(s): Igor Vivanco, King's College London and Chiara Palladino, F. Hoffmann-La Roche AG

Review Timeline:

Submission Date:	2021-09-16
Editorial Decision:	2021-10-20
Revision Received:	2022-03-25
Editorial Decision:	2022-04-19
Revision Received:	2022-05-04
Accepted:	2022-05-05

Transaction Report:

October 20, 2021

Re: Life Science Alliance manuscript #LSA-2021-01235-T

Igor Vivanco
Institute of Pharmaceutical Sciences, King's College London
United Kingdom

Dear Dr. Vivanco,

Thank you for submitting your manuscript entitled "DUSP4 protects BRAF and NRAS-mutant melanoma cells from oncogene overdose through modulation of MITF function" to Life Science Alliance. The manuscript was assessed by expert reviewers, whose comments are appended to this letter. We invite you to submit a revised manuscript addressing the Reviewer comments.

Thank you for this interesting contribution to Life Science Alliance. We are looking forward to receiving your revised manuscript.

Sincerely,

B. MANUSCRIPT ORGANIZATION AND FORMATTING:

Reviewer #1 (Comments to the Authors (Required)):

Melanoma represents a prime example for an ERK/MAPK pathway addicted tumour entity as the overwhelming majority carries mutations in either KIT, NRAS or BRAF proto-oncogenes. In addition to the more established and the about twenty-year old concept of addiction to high ERK pathway activity, there is accumulating evidence that, depending on the (epi)genetic context of the oncogene expressing cell, high intensity ERK signals can also trigger oncogene induced senescence or oncogene overdose. The latter has been more recently described upon drug withdrawal or upon introduction of another oncogenic lesion in the same pathway that is usually mutually exclusive to the initial alteration. Thus, the oncogene overdose phenomenon represents an interesting and at first sight sometimes counterintuitive concept that could explain certain observations with targeted therapy compounds and could be potentially exploited during so-called drug holidays. Consequently, a deeper understanding of the molecular mechanism underlying this oncogene overdose concept could help to tailor targeted therapy concepts promising fewer side-effects, and smarter combination therapies that break drug resistance. Indeed, secondary or acquired drug resistance to clinically relevant BRAF and MEK inhibitors is a frequent limitation of an initially successful targeted therapy, not only in melanoma but also in other ERK pathway driven neoplasia. Interestingly, however, drug resistant ERK addicted tumours often remain ERK driven as they have found ways to reactivate or maintain pathway activity despite the presence of the inhibitor. Upon sudden drug withdrawal, these tumours then often react with cell cycle arrest or cell death caused by an overshoot in ERK pathway activity. Thus, the identification of the mechanisms fine-tuning ERK activity could highlight novel critical targets (or "brakes"), whose loss-of-function could synergise with resurging ERK pathway input upon withdrawal of RAF or MEK inhibitors. In the present manuscript, the authors approached this highly timely and clinically relevant task by conducting a focused siRNA based screen in BRAF V600E driven melanoma lines on potential negative ERK pathway regulators, such as dual specificity phosphatases (DUSP), which often but not exclusively counteract the action of MEKs by dephosphorylating the activation loop of ERKs, other phosphatases but also negative regulators, such as SPRED or Sprouty, which intercept with RTK and/or RAF activation. Stimulated by the outcome of the screen in SKMEL28 cells, the authors observed that depletion of DUSP4, a phosphatase known to prefer ERK as a substrate over the other MAPK, conferred a very robust cell growth inhibition and cell death induction in three other melanoma lines. They confirm previous findings that DUSP4 acts primarily on the ERK axis and that its RNAi mediated depletion further increases ERK phosphorylation. This part of the manuscript is well-conducted and delivers interesting, albeit not completely unanticipated results, which are still relevant as DUSP4 has been implicated as a modulator of MEK inhibitor sensitivity previously. Very interesting is the observation that ERK hyper-activation triggered by DUSP4 loss is associated with the downregulation of lineage-defining genes, including that for the lineage- and differentiation defining transcription factor MITF. They also show that the degree of MITF expression in ERK pathway inhibitor resistant cells (generated by exposure to high trametinib or vemurafenib doses) determines their requirement for DUSP4 expression. This is an interesting finding also from the angle of tumour heterogeneity and therapy induced phenotypic plasticity. In summary, the manuscript by Gutierrez-Prat provides several stimulating findings and experimental evidence for the emerging concept of oncogene overdose toxicity. The manuscript is well-written and most data are clearly presented. Another strength is that key observations are reproduced in several cell lines and/or isogenic models. A few aspects, however, should be addressed in a revised version.

Major:

1. The Western blots for some key data in certain experiments, e.g. MITF and pERK levels in Fig. 4A, Fig. 5E-I, Supplementary Figure 3C/E, Supplementary Figure 5 (in particular the A375 panel), should be substantiated with quantification, preferably from three independent biological replicates. In some panels, e.g. the MITF detection in A375 lysates in Supplementary Figure 5, it is possible that small transfer artifacts (lane 3, siDUSP6) are present.
2. Colony forming assay in Supplementary Figure 6C. Were really 1000 micromolar Dabrafenib used? Or should it rather read "nM"?
3. The authors should briefly discuss their finding in context with previous studies implicating MITF as an important modulator of drug sensitivity in melanoma (see PMID: 28976960 for example).
4. The observation that MITF expression was not altered by DUSP6 and DUSP10 expression is interesting as, at least DUSP6 follows a similar spatio-temporal expression pattern. Could the authors briefly speculate as to why this might be the case? Is it because DUSP4 is primarily a nuclear DUSP, while DUSP6 is a typical cytoplasmic DUSP and DUSP10 is both (PMID: 17473844)? Moreover, the depletion efficiency for DUSP6 and DUSP10 should be provided for Supplementary Figure 4B. At least for DUSP6 very specific antibodies are commercially available.

Minor:

5. Please provide page numbers! This facilitates reading and review drafting!

Reviewer #2 (Comments to the Authors (Required)):

In this manuscript, Gutierrez-Prat et al. describe their discovery that the absence of DUSP4 from MITF-positive melanoma cells leads to hyperactivation of the MAPK pathway and apoptosis. They show this to be the case in both drug-naïve and resistant cells. This identifies DUSP4 as a potential drug target in melanoma. The paper is clearly written, the data is convincingly presented and the figures clearly illuminate the results. The link between DUSP4 and the transcription of MITF is the least clear aspect of the manuscript. First, it is not clear if the MITF-knockout cells die by apoptosis like the DUSP4 knockdown cells or if they become non-proliferative (more likely). Second, it is not clear if overexpression of MITF or PAX3 rescue the effects of DUSP4 knockdown. Third, according to the RNA-seq data DUSP4 seems to reduce the expression of both MITF and PAX3 but the mechanism is not entirely clear.

Minor points:

1. In several places the authors discuss „cell toxicity" where they have really shown induction of apoptosis.
2. Two references are listed as PID numbers.
3. The use of Skmel28 and A375 cells as low and high MITF cells is logical. However, due to the dissimilar genetic backgrounds of these cells it would have been better to use Skmel28 lacking MITF as a control for the Skmel28 cells.

Reviewer #3 (Comments to the Authors (Required)):

It has been shown that BRAF or NRAS-mutant melanoma cells acquire resistance to MAPKi almost invariably, becoming addicted to the inhibitor. When this happens, the cells require drug exposure to maintain a residual level of signaling and proliferate. Acute drug withdrawal provokes oncogene overdose which displays the same outcome as oncogene inhibition. In this scenario, drug holidays are proposed as a therapeutic strategy, but this is only valid for inhibitor-resistant and not for treatment-naïve cells. The main finding of this manuscript is that DUSP4 depletion leads to toxic ERK activation levels in both, drug-naïve and drug-resistant melanoma cells and these effects are independent of the oncogenic background (NRAS or BRAF mutations). Additionally, the authors found that DUSP4 is only relevant when the cells express certain amounts of MITF, from moderate to high, and not when it is absent. Hence, the authors propose DUSP4 as a therapeutic target for a subset of melanomas based on MITF expression, independent of the oncodriver mutation or whether they are treatment-naïve or resistant.

Point 1. With respect to the statement "Loss of the MAPK phosphatase DUSP4 is deleterious to BRAFV600E melanoma cells", the authors clearly demonstrated a negative effect of DUSP4 knockdown in cell growth and proliferation and an increase in apoptosis. However, there are a few points to take into consideration.

The siRNA was carried out only in one melanoma cell line: SKMEL28 (F1A and S1), and apparently, only once. If N is different than 1 indicate this and add error bars. They show the effect observed in growth inhibition (cell confluence) upon siRNA transfection, but an immunoblot or mRNA analysis would demonstrate that the siRNA actually worked.

Downregulation of PEA15 has a similar effect to inhibit cell growth as does suppression of DUSP4, but the authors do not highlight this finding. PEA15 is a cytoplasmic anchor of ERK. Is ERK localization a main role here as well as ERK dephosphorylation by DUSP4? Even if the authors did not further investigate the role of PEA15, the strong negative effect of the downregulation of this protein should not be ignored.

The authors validated the DUSP4 downregulation effect in another melanoma cell line (COLO829), and compared it to the downregulation of DUSP6 and DUSP10 which are other negative regulators with substrate specificity that overlaps with that of DUSP4. They observed that DUSP4 downregulation provokes DUSP6 and DUSP10 upregulation, but this does not compensate the growth inhibitory effects of DUSP4 loss. They state that DUSP4 increases DUSP6 and DUSP10 mRNA and protein, but no immunoblots for the proteins were shown. DUSP6 and DUSP10 blots would substantiate the idea that the lack of compensation is not due to a lack of mRNA translation into protein.

Point 2. With respect to the statement "DUSP4 downregulation induces oncogenic overdose through ERK overactivation in mutant melanoma cells", the authors demonstrate that sub-therapeutic Trametinib doses reverse the effect of DUSP4 depletion in SKMEL28 and COLO829, and this is mediated by ERK specifically. Are COLO829 cells MITF-high as is SKMEL28?

Point 3. Concerning the statement "DUSP4-mediated ERK activity controls the expression of MITF and its target genes in a lineage dependent fashion", the authors show that specifically depletion of DUSP4, and not DUSP6 or DUSP10, downregulates the expression of MITF target genes and attribute the effect to ERK because Trametinib sub-lethal doses rescue the effects of DUSP4 depletion. They provide additional evidence that DUSP4 downregulation effects are specific to BRAF-mutant melanoma cells with high MITF levels by showing that DUSP4 knock down has no effect in BRAF mutant tumors from other tissues, suggesting that DUSP4 is only relevant in the presence of MITF. Yet how is it that they do not observe any impact of BRAF

downregulation on the amount of pERK in the BRAF-mutant tumors from other tissues (S4C), but they do see a decrease in growth (F3F). Are these other cell types dependent on the BRAF mutation or not?

Point 4. They state that "the essential role of DUSP4 is restricted to MITF-expressing melanoma and it is independent of the oncogene mutation". There are NRAS and BRAF melanomas with varying MITF expression independent of the oncogene mutation. The authors demonstrated that DUSP4 depletion increases pERK either in MITF-high or low melanoma cells, but the effect on cell growth is exclusive to MITF-high cells. This suggests that the effects of DUSP4 depletion depend on MITF rather than an optimal level of ERK activity alone. To support their conclusion, the authors used two MITF-high and two MITF-low melanoma cell lines. For consistency, it would be useful to demonstrate that these effects are specific to DUSP4 and not DUSP6 or DUSP10 in SKMEL30 (MITF-high) (S5).

Point 5. With respect to the hypothesis that MITF expression levels in treatment-naïve cells determine the essentiality of DUSP4 in MAPKi resistant cells, the authors generated SKMEL28 (MITF-high) and A375 (MITF-low) resistant cells with chronic exposure to lethal doses of Trametinib or Vemurafenib, respectively. In SKMEL28 cells an acute, but not sustained, treatment withdrawal strongly compromised cell viability, but this was not the case for A375 cells with low MITF expression and without variations in DUSP4. It would be clearer if cell viability were shown in all the conditions (Parental, R+T, R-T sustained and R-T acute for SKMEL28; Parental, R+T, and R-T acute for A375) by CellTiter-Glo, as in figure S6A/B.

The authors state that "these results suggest that the role of DUSP4 is solely critical in high-MITF derived cells with acquired resistance to MAPKi". However, in F5B, they observe a loss of DUSP4 expression and, in consequence, MITF in R+T SKMEL28 cells. In this condition the cells are viable even though there is no DUSP4 to protect them. The authors justify this fact in the discussion suggesting that the MAPKi might be suppressing DUSP4 expression to compensate for the lack of signal.

Nevertheless, DUSP4 should be essential for survival, what is the explanation for these cells being perfectly viable without DUSP4, without MITF, and with hyperactivated ERK if the Trametinib dose is a lethal dose instead of the sub-therapeutic dose that would normalize pERK levels?

Minor

1. There is a typo in S2E and F figure legend, shDUSP4-1 instead of shDUSP4-2.
2. In the section on possible mechanisms of acquired resistance, a previous study (Reference 35) was cited but the sentence is confusing and might be improved by inserting "it was previously reported".
3. The figure numbers are mixed up in the statement on MITF-independent mechanisms (Figure 5B, 5E and 5H). 5H refers to A375 cells.

Reviewer #1*Comments:*

Melanoma represents a prime example for an ERK/MAPK pathway addicted tumour entity as the overwhelming majority carries mutations in either KIT, NRAS or BRAF proto-oncogenes. In addition to the more established and the about twenty-year old concept of addiction to high ERK pathway activity, there is accumulating evidence that, depending on the (epi)genetic context of the oncogene expressing cell, high intensity ERK signals can also trigger oncogene induced senescence or oncogene overdose. The latter has been more recently described upon drug withdrawal or upon introduction of another oncogenic lesion in the same pathway that is usually mutually exclusive to the initial alteration. Thus, the oncogene overdose phenomenon represents an interesting and at first sight sometimes counterintuitive concept that could explain certain observations with targeted therapy compounds and could be potentially exploited during so-called drug holidays. Consequently, a deeper understanding of the molecular mechanism underlying this oncogene overdose concept could help to tailor targeted therapy concepts promising fewer side-effects, and smarter combination therapies that break drug resistance. Indeed, secondary or acquired drug resistance to clinically relevant BRAF and MEK inhibitors is a frequent limitation of an initially successful targeted therapy, not only in melanoma but also in other ERK pathway driven neoplasia. Interestingly, however, drug resistant ERK addicted tumours often remain ERK driven as they have found ways to reactivate or maintain pathway activity despite the presence of the inhibitor. Upon sudden drug withdrawal, these tumours then often react with cell cycle arrest or cell death caused by an overshoot in ERK pathway activity. Thus, the identification of the mechanisms fine-tuning ERK activity could highlight novel critical targets (or “brakes”), whose loss-of-function could synergise with resurging ERK pathway input upon withdrawal of RAF or MEK inhibitors. In the present manuscript, the authors approached this highly timely and clinically relevant task by conducting a focused siRNA based screen in BRAF V600E driven melanoma lines on potential negative ERK pathway regulators, such as dual specificity phosphatases (DUSP), which often but not exclusively counteract the action of MEKs by dephosphorylating the activation loop of ERKs, other phosphatases but also negative regulators, such as SPRED or Sprouty, which intercept with RTK and/or RAF activation. Stimulated by the outcome of the screen in SKMEL28 cells, the authors observed that depletion of DUSP4, a phosphatase known to prefer ERK as a substrate over the other MAPK, conferred a very robust cell growth inhibition and cell death induction in three other melanoma lines. They confirm previous findings that DUSP4 acts primarily on the ERK axis and that its RNAi mediated depletion further increases ERK phosphorylation. This part of the manuscript is well-conducted and delivers interesting, albeit not completely unanticipated results, which are still relevant as DUSP4 has been implicated as a modulator of MEK inhibitor sensitivity previously. Very interesting is the observation that ERK hyper-activation triggered by DUSP4 loss is associated with the downregulation of lineage-defining genes, including that for the lineage- and differentiation defining transcription factor MITF. They also show that the degree of MITF expression in ERK pathway inhibitor resistant cells (generated by exposure to high trametinib or vemurafenib doses) determines their requirement for DUSP4 expression. This is an interesting finding also from the angle of tumour heterogeneity and therapy induced phenotypic plasticity.

In summary, the manuscript by Gutierrez-Prat provides several stimulating findings and experimental evidence for the emerging concept of oncogene overdose toxicity. The manuscript is well-written and most data are clearly presented. Another strength is that key observations are reproduced in several cell lines and/or isogenic models. A few aspects, however, should be

addressed in a revised version.

Major:

1. The Western blots for some key data in certain experiments, e.g. MITF and pERK levels in Fig. 4A, Fig. 5E-I, Supplementary Figure 3C/E, Supplementary Figure 5 (in particular the A375 panel), should be substantiated with quantification, preferably from three independent biological replicates. In some panels, e.g. the MITF detection in A375 lysates in Supplementary Figure 5, it is possible that small transfer artifacts (lane 3, siDUSP6) are present.

We thank the reviewer for highlighting the timeliness and clinical relevance of our study as well as the broadly positive assessment of our experimental approach and results.

We thank the reviewer for pointing out the need to quantify some of our western blot data. To address this, we have included quantifications of MITF and P-ERK protein levels in requested experiments. We have also made clear in the figure legends that the experiments were performed three independent times. The experiments in the original Supplementary Figure 5 have been repeated and replaced including additional cell lines (SKMEL30) and siDUSP10 alone or in combination with siDUSP4 (new Supplemental Figure 6). In the repeat experiment, we confirm that the MITF levels from A375 cells are indeed very low compared to SKMEL28 and SKMEL30 cell lines.

2. Colony forming assay in Supplementary Figure 6C. Were really 1000 micromolar Dabrafenib used? Or should it rather read “nM”?

We apologise for this typographical error. We have relabeled the colony formation assay (new Supplementary Figure 7C) with the correct units (nanomolar) of Dabrafenib concentration.

3. The authors should briefly discuss their finding in context with previous studies implicating MITF as an important modulator of drug sensitivity in melanoma (see PMID: 28976960 for example).

We thank the reviewer for the excellent suggestion. We have discussed our findings in the context of previous studies considering the role MITF in regulating drug sensitivity in melanoma. This has been added to the discussion section of the revised manuscript (p13 of the revised manuscript).

4. The observation that MITF expression was not altered by DUSP6 and DUSP10 expression is interesting as, at least DUSP6 follows a similar spatio-temporal expression pattern. Could the authors briefly speculate as to why this might be the case? Is it because DUSP4 is primarily a nuclear DUSP, while DUSP6 is a typical cytoplasmic DUSP and DUSP10 is both (PMID: 17473844)? Moreover, the depletion efficiency for DUSP6 and DUSP10 should be provided for Supplementary Figure 4B. At least for DUSP6 very specific antibodies are commercially available.

We thank the reviewer for this insightful comment. As the reviewer points out, downregulation of either DUSP6 or DUSP10 is insufficient to modify P-ERK levels and therefore MITF expression is unaffected by these perturbations (Supplementary Figure 4B). As suggested by the referee, this

could be due to differences in the subcellular distribution of DUSPs (Huang & Tan, 2012) (Jeffrey, Camps et al., 2007). DUSP4 is a nuclear phosphatase which has been shown to prevent activation of ERK in the nucleus (Cagnol & Rivard, 2013). In contrast, DUSP6 appears to be primarily cytoplasmic (Karlsson, Mathers et al., 2004), while DUSP10 has a mixed distribution (Tanoue, Moriguchi et al., 1999). Interestingly, one of the other hits in our RNAi screen was PEA15, a scaffolding protein that serves as a cytoplasmic anchor for ERK and prevents its nuclear localization (Formstecher, Ramos et al., 2001). This suggests that perhaps the selective sensitivity to DUSP4 depletion may be associated with specific superactivation of ERK in the nucleus. Additionally, our analysis of publicly available gene expression data shows that DUSP4 is the most highly expressed isoform among melanoma cells, compared to cancer cell lines derived from other tissues (Rebuttal Figure 1A). Consistently, when we assessed the expression of DUSPs in COLO829 and SKMEL28 cells, DUSP4 expression was highest compared to DUSP6 and DUSP10 (Rebuttal Figure 1B), suggesting that DUSP4 function might be most relevant in the melanoma setting. Of note, the expression of DUSP6 and DUSP10 is upregulated upon DUSP4 knockdown, most likely due to a compensatory effect. However, despite their enhanced expression, the levels of P-ERK are still notably increased (Supplementary Figure 4B). These observations suggest that compared to DUSP6 and DUSP10, DUSP4 may have a more significant role in the specific dephosphorylation of ERK (compared to other MAPKs). Additional text has been added to the discussion to cover these points more explicitly (p12 of the revised manuscript).

The appropriate dilution of DUSP6 antibody was optimized for COLO829 and SKMEL28 cell lines, and western blots have been added (Supplementary Figure 4B). However, as the reviewer seems to have suggested, the quality of the DUSP10 antibody did not seem to be ideal and western blots did not work well in these cells. Instead, its depletion efficiency was assessed at the mRNA level (Supplementary Figure 4C).

A

B

Rebuttal Figure 1. Analysis of DUSP4, DUSP6 and DUSP10 mRNA expression in melanoma cell lines. (A) Gene expression data was obtained from the public available Expression Atlas database (<http://www.ebi.ac.uk/gxa>) (Papatheodorou, Fonseca et al., 2018). In the upper panel, 39 melanoma cell lines are shown whereas in the lower panel 12 cell lines from different tissues are represented. The expression of each gene is represented per transcripts per million (TPM). Genes are labelled with different colours. (B) mRNA from SKMEL28 and COLO829 melanoma cells was extracted and gene expression of DUSP4, DUSP6 and DUSP10 is shown.

Minor:

5. Please provide page numbers! This facilitates reading and review drafting!

We apologise for the oversight. Page numbers have been added.

Reviewer #2

Comments:

In this manuscript, Gutierrez-Prat et al. describe their discovery that the absence of DUSP4 from MITF-positive melanoma cells leads to hyperactivation of the MAPk pathway and apoptosis. They show this to be the case in both drug-naive and resistant cells. This identifies DUSP4 as a potential drug target in melanoma. The paper is clearly written, the data is convincingly presented and the figures clearly illuminate the results.

The link between DUSP4 and the transcription of MITF is the least clear aspect of the manuscript.

First, it is not clear if the MITF-knockout cells die by apoptosis like the DUSP4 knockdown cells or if they become non-proliferative (more likely).

We thank the reviewer for bringing this important point to our attention. We have addressed the question by knocking down MITF in SKMEL28 cells and analyzing cell proliferation and cell death with Cell tracer and Annexin V staining, respectively (Rebuttal Figure 2). Consistent with previous reports (Dilshat, Fock et al., 2021), MITF knockdown (Rebuttal Figure 2A) leads to significant proliferation arrest as determined by phase-contrast images and Cell Tracer Violet dilution assays (Rebuttal Figure 2B-2C) (Dilshat et al., 2021). However, as suggested by the Reviewer, compared to DUSP4 silencing, MITF knockdown showed only a slight (albeit consistent) effect on cell death as measured by Annexin V staining (Rebuttal Figure 2D). In fact, the magnitude of the cytotoxic effect seen with DUSP4 knockdown is generally lower compared to the more robust cytostatic effect observed across relevant models tested (Manuscript Figure 1). These results strongly suggest that MITF downregulation is sufficient to elicit the consistent cytostatic response associated with DUSP4 depletion, but that the effects on cell death might involve additional factors. These data has now been added as Supplementary Figure 5 and has been included in the text (p7 of the revised manuscript).

Rebuttal Figure 2. Analysis of cell proliferation and cell death upon MITF knockdown. (A) SKMEL28 were transfected with siRNA against MITF (siMITF) or non-targeting control (siNT). 48h later, cell lysates were analyzed by western blot. Band intensities were quantified by ImageJ software and MITF/GAPDH ratio is indicated of the blot shown. (B) Phase-contrast images are shown depicting cell confluence and morphology 7 days after the transfection. (C) Transfected cells were stained with Cell Tracer Violet (CTBV) and analyzed by CTBV incorporation the same day of the staining (Day0) or after 7 days (Day7). The CTBV incorporation at Day7 is shown. Histogram indicates Mean Fluorescence Intensity (MFI) of siNT and siMITF transfected cells at Day7. Data represents mean \pm SEM of four technical replicates. (D) Cell death was assayed by Annexin V and Zombie stainings. Early apoptosis indicates the percentage of Annexin V⁺/Zombie⁻ cells whereas late apoptosis shows the percentage of Annexin V⁺/Zombie⁺ cells. Data represents mean \pm SEM of four technical replicates.

Second, it is not clear if overexpression of MITF or PAX3 rescue the effects of DUSP4 knockdown.

We thank the reviewer for raising this important point. MITF is a key transcription factor for melanocytes and regulates cell cycle progression, survival and differentiation. To explain the paradoxical observation that MITF could both promote and inhibit proliferation of melanoma cells, it has been proposed that the level of MITF activity is a determinant of phenotype switching in melanoma cells. According to this “MITF rheostat” model, high levels of MITF are associated with cell differentiation and reduced proliferation, while progressively decreasing MITF levels are associated with proliferation, dedifferentiation/ invasion, senescence, and eventually cell

death (Goding, 2011, Goding & Arnheiter, 2019, Seberg, Van Otterloo et al., 2017). Therefore, the levels of MITF are tightly regulated in melanoma cells leading to different cell states. Consistent with this notion, when we tried to overexpress MITF in DUSP4 KD SKMEL28 cells in order to perform a rescue experiment, most of the cells died after transfection and the attached/viable cells did not overexpress MITF (Rebuttal Figure 3). We suspect that since SKMEL28 already express high levels of MITF, further overexpression likely induces terminal differentiation and cell cycle arrest as reported in other studies (Loercher, Tank et al., 2005).

Third, according to the RNA-seq data DUSP4 seems to reduce the expression of both MITF and PAX3 but the mechanism is not entirely clear.

We thank the reviewer for the keen observation and apologise for our lack of clarity on this point. MITF expression is regulated by several transcription factors, most prominently SOX10 and PAX3 (Wellbrock & Arozarena, 2015). Our RNAseq data show that DUSP4 depletion leads to a decrease in MITF transcript levels, while the levels of both SOX10 and PAX3 were not significantly altered (Manuscript Figure 3A and Rebuttal Figure 3C). However, we also analysed these data using Virtual Inference of Protein-activity by Enriched Regulon (VIPER) (Alvarez, Shen et al., 2016), an algorithm that allows computational inference of protein activity by using gene expression of target genes as a reporter of transcription factor activity. Through this analysis we find that PAX3 transcriptional activity (rather than its expression levels) is repressed in DUSP4 knockdown cells and rescued upon trametinib treatment (Manuscript Figure 3C). Accordingly, trametinib treatment also reestablished the expression levels of MITF as well as cell viability in DUSP4 knockdown cells (Manuscript Figure 3C and Figure 2C). Together, these results suggest that the enhanced P-ERK levels observed in the absence of DUSP4, negatively regulate PAX3 activity and consequently, MITF expression in melanoma cells (Manuscript Figure 3). Negative regulation of MITF through ERK-PAX3 signaling has been also supported by other studies (Smith, Rana et al., 2019). We have added some additional text in the manuscript to clarify this point (p7 of the revised manuscript).

Rebuttal Figure 3. Regulation of MITF in melanoma cells. (A-B) Transient transfection of MITF and GFP as control was performed in SKMEL28 cells. After 48h, pictures of all conditions were taken and representative images are shown (A). Cell lysates were analyzed by western blot (B). (C) SKMEL28 and COLO829 cells were transfected with siRNAs against DUSP4 and non-targeting as control. After 48h, mRNA was isolated and PAX3 and SOX10 genes were analyzed by RT-qPCR.

Minor points:

1. In several places the authors discuss „cell toxicity" where they have really shown induction of apoptosis.

We apologise for the confusing terminology. We have now replaced “cell toxicity” with “cell death”.

2. Two references are listed as PID numbers.

We apologise for this oversight. The two PMID numbers have been replaced with proper bibliographic references.

3. The use of Skmel28 and A375 cells as low and high MITF cells is logical. However, due to the dissimilar genetic backgrounds of these cells it would have been better to use Skmel28 lacking MITF as a control for the Skmel28 cells.

We thank the reviewer for this insightful comment and agree that an isogenic cell line pair might have provided a more suitable control (Goding, 2011, Goding & Arnheiter, 2019). However, as shown in Manuscript Figure 4B and also as shown by others (Dilshat et al., 2021), MITF depletion in SKMEL28 causes severe proliferative defects which renders these cells less than ideal for further experimentation. We also considered the generation of stable knockdown clones with better proliferative capacity, but thought that this would take a long time to generate and introduce significant bias. Additionally, we unintentionally generated MITF-deficient SKMEL28 cells when we made MAPKi-resistant cells. We find that chronic exposure to therapeutic concentrations of trametinib led to the emergence of drug-resistant clones (SKMEL28 R+ T) characterised by MITF loss (Manuscript Figure 5B), and trametinib addiction (Supplementary Figure 7C). When we subjected these cells to long-term trametinib withdrawal, the resistant cells were not only still MITF-deficient, but also remained sensitive to DUSP4 depletion possibly due to their acquired mesenchymal features and RTK activation profile (Figure 5C and 5E). These data suggest that the MITF status at the onset of tumorigenesis may be the key determinant of DUSP4 dependence.

Reviewer #3

Comments:

It has been shown that BRAF or NRAS-mutant melanoma cells acquire resistance to MAPKi almost invariably, becoming addicted to the inhibitor. When this happens, the cells require drug exposure to maintain a residual level of signaling and proliferate. Acute drug withdrawal provokes oncogene overdose which displays the same outcome as oncogene inhibition. In this scenario, drug holidays are proposed as a therapeutic strategy, but this is only valid for inhibitor-resistant and not for treatment-naïve cells. The main finding of this manuscript is that DUSP4 depletion leads to toxic ERK activation levels in both, drug-naïve and drug-resistant melanoma cells and these effects are independent of the oncogenic background (NRAS or BRAF mutations). Additionally, the authors found that DUSP4 is only relevant when the cells express certain amounts of MITF, from moderate to high, and not when it is absent. Hence, the authors propose DUSP4 as a therapeutic target for a subset of melanomas based on MITF expression, independent of the oncodriver mutation or whether they are treatment-naïve or resistant.

Point 1. *With respect to the statement "Loss of the MAPK phosphatase DUSP4 is deleterious to BRAFV600E melanoma cells", the authors clearly demonstrated a negative effect of DUSP4 knockdown in cell growth and proliferation and an increase in apoptosis. However, there are a few points to take into consideration. The siRNA was carried out only in one melanoma cell line: SKMEL28 (F1A and S1), and apparently, only once. If N is different than 1 indicate this and add error bars. They show the effect observed in growth inhibition (cell confluence) upon siRNA transfection, but an immunoblot or mRNA analysis would demonstrate that the siRNA actually worked.*

We thank the reviewer for the comments, and apologise for the rationale and methodology not being presented more clearly. In order to identify either direct or indirect negative regulators of the BRAF-MEK-ERK signaling with a potential effect on cell viability, we performed a siRNA-based screen considering known regulators of the MAPK pathway. As a primary screen, every

gene was measured only once in one representative melanoma cell line (SKMEL28). The intent was simply to generate hits (i.e. a hypothesis) that could be further validated in subsequent experiments using additional models. However, we did include three technical replicates in the analysis. The values that were previously shown in the graph represented the average of all three technical replicates. In the revised version of the Manuscript, we have replaced the average data with the values of every single replicate to visualize the variability among technical replicates (Manuscript Figure 1A and Supplementary Figure 1). Since the primary screening included more than 40 genes, we did not evaluate the efficacy of each single siRNA. Instead, we selected the gene candidates with the strongest cell growth inhibition and we validated their downregulation as well as their contribution in cell growth in different melanoma cell lines (Manuscript Figure 1A and Supplementary Figure 1).

Downregulation of PEA15 has a similar effect to inhibit cell growth as does suppression of DUSP4, but the authors do not highlight this finding. PEA15 is a cytoplasmic anchor of ERK. Is ERK localization a main role here as well as ERK dephosphorylation by DUSP4? Even if the authors did not further investigate the role of PEA15, the strong negative effect of the downregulation of this protein should not be ignored.

We thank the reviewer for pointing this out, and apologise for failing to comment on PEA15. As noted by the reviewer, downregulation of both DUSP4 and PEA15 led to significant cell growth inhibition in our RNAi screen. However, DUSP4 downregulation showed a consistent cell growth impairment among all four melanoma cell lines tested (Manuscript Figure 1 and Rebuttal Figure 4), whereas depletion of PEA15 only inhibited growth in 3 out of the 4 models (Rebuttal Figure 4). Since PEA15 acts as a cytoplasmic anchor for ERK, and given that ERK is constitutively activated in NRAS and BRAF mutant melanoma cells, it is entirely possible that melanoma cells are sensitive to PEA15 depletion due to nuclear ERK overdose. In fact, DUSP4 is a nuclear phosphatase, and therefore, it is possible that the growth inhibitory effects of DUSP4 and PEA15 knockdown have a similar mechanistic basis. However, we feel that fully validating this additional hypothesis would be beyond the scope of this work. Nevertheless, we have added PEA15 to our discussion to highlight this possibility. We, instead, decided to focus our follow-up work on DUSP4 for two reasons. First, we saw a very consistent phenotype in all melanoma cells analyzed, which is consistent with what has been observed in genome-wide CRISPR-Cas9 genetic screens from several human mutant melanoma cell lines (<https://depmap.org/ceres/>). And second, we considered the potential feasibility of developing targeted therapeutic agents against our hits in the future, and the consensus in the drug development field is that enzymes represent a more drugable target class compared to scaffolding proteins. We have added some text in the discussion to clarify some of these points (p12 of the revised manuscript).

Rebuttal Figure 4. SKMEL28, COLO829, RVH421 and SKMEL24 melanoma cells were transfected with indicated siRNAs against DUSP4, PEA15 or with a non-targeting control. Graphs show the growth curves of transfected cells by measuring cell confluence over time. Cell growth values were normalized against time 0. After 48 post-transfection, cell lysates were analyzed by western blot.

The authors validated the DUSP4 downregulation effect in another melanoma cell line (COLO829), and compared it to the downregulation of DUSP6 and DUSP10 which are other negative regulators with substrate specificity that overlaps with that of DUSP4. They observed that DUSP4 downregulation provokes DUSP6 and DUSP10 upregulation, but this does not compensate the growth inhibitory effects of DUSP4 loss. They state that DUSP4 increases DUSP6 and DUSP10 mRNA and protein, but no immunoblots for the proteins were shown. DUSP6 and DUSP10 blots would substantiate the idea that the lack of compensation is not due to a lack of mRNA translation into protein.

We thank the reviewer for bringing this up. We have indeed evaluated the expression levels of DUSP6 and DUSP10 following DUSP4 knockdown at both mRNA and protein level. While DUSP6 antibodies were of sufficient quality to produce reliable western blot data, DUSP10 protein levels were difficult to detect with commercially available reagents (as was alluded to by reviewer #1), and its expression was analyzed at the mRNA level instead (Supplementary Figure 2B, 4C, and 6B). Additionally, we find that knockdown of DUSP10 does not significantly affect cell viability or ERK phosphorylation in all models tested (Manuscript Figure 1B and Supplementary Figures 6A and 6C), suggesting that in this context, ERK is unlikely to be the preferred substrate of DUSP10. We therefore believe that while we cannot formally rule out the possibility of DUSP10 protein not being upregulated by DUSP4 knockdown, cells that are sensitive to DUSP4 knockdown are ultimately unable to prevent the superactivation of ERK. In addition to adding DUSP6 western blot data (Supplementary Figures 4B and 6A), we have taken additional care to qualify our statements about DUSP10 (p4 of the revised manuscript).

Point 2. *With respect to the statement "DUSP4 downregulation induces oncogenic overdose through ERK overactivation in mutant melanoma cells", the authors demonstrate that sub-therapeutic Trametinib doses reverse the effect of DUSP4 depletion in SKMEL28 and COLO829, and this is mediated by ERK specifically. Are COLO829 cells MITF-high as is SKMEL28?*

We thank the reviewer for raising this point, and apologise for not being sufficiently explicit about MITF status in these cells. SKMEL28 and COLO829 cells indeed have similarly high protein levels of MITF. A western blot documenting this has now been added as Supplementary Figure 4E. To further elaborate on this point, we have carried out additional qPCR analysis of MITF target genes in both SKMEL28 and COLO829 and their response to DUSP4 perturbation (This has been added as Supplementary Figure 4A). Similar to SKMEL28, we find that the observed decrease in MITF levels following DUSP4 knockdown in COLO829 is associated with a decrease in MITF target genes, an effect that is reversed by treatment with sub-therapeutic doses of trametinib (Manuscript Figure 3D and Supplementary Figure 4A and 4B). This suggests that MITF is fully functional in COLO829 cells, and that DUSP4 is a critical regulator of its transcriptional activity. Therefore, we consider both COLO829 and SKMEL28 to be MITF-high expressing cell lines.

Point 3. *Concerning the statement "DUSP4-mediated ERK activity controls the expression of MITF and its target genes in a lineage dependent fashion", the authors show that specifically depletion of DUSP4, and not DUSP6 or DUSP10, downregulates the expression of MITF target genes and attribute the effect to ERK because Trametinib sub-lethal doses rescue the effects of DUSP4 depletion. They provide additional evidence that DUSP4 downregulation effects are specific to BRAF-mutant melanoma cells with high MITF levels by showing that DUSP4 knock down has no effect in BRAF mutant tumors from other tissues, suggesting that DUSP4 is only relevant in the presence of MITF. Yet how is it that they do not observe any impact of BRAF downregulation on the amount of pERK in the BRAF-mutant tumors from other tissues (S4C), but they do see a decrease in growth (F3F). Are these other cell types dependent on the BRAF mutation or not?*

We thank the reviewer for bringing this to our attention, and fully agree that this is an important point to address. We originally provided data on two BRAF-mutant non-melanoma cell lines, namely DBTRG (a glioma cell line) and HT29 (a colorectal carcinoma cell line). We suspected that the inability of BRAF knockdown to cause a significant change in ERK phosphorylation in these lines was the result of both non-technical and technical issues. Firstly, there is ample evidence that in BRAF-mutant non-melanoma tumours that do not respond to single agent BRAF inhibitors (e.g. colorectal carcinoma and thyroid cancer), there is release of negative feedback signals upstream (often involving receptor tyrosine kinases) of MAPK that cause pathway reactivation and de novo drug resistance. This is in contrast to BRAF-inhibitor-responsive melanoma cells where such negative feedback release is much less common. This raises the possibility that cell-type-specific negative feedback networks will have an impact on a) the activation state of the pathway in response to various perturbations, and b) the extent by which these perturbation affect the viability of these cells. However, this layer of regulation may not necessarily affect the underlying dependence of these cells on the function of BRAF. Secondly, the kinetics of phosphorylation/dephosphorylation and turnover rates of active vs inactive MAPKs could influence the pERK readout given the amount of time it takes to downregulate BRAF following siRNA knockdown and when the cells are collected. However, this encouraged us to inspect the literature to assess the experience of other groups with the use of these particular cell lines. We found that while DBTRG is consistently reported as BRAF-dependent and exhibits MAPK pathway activation that is sensitive to BRAF inhibitors, there are some inconsistencies about HT29 sensitivity to BRAF inhibition with some groups not considering this cell line completely BRAF dependent (Mao, Tian et al., 2013),(Su, Zhan et al., 2021), (Ahmed, Adamopoulos et al., 2019). We therefore looked for another BRAF-mutant colorectal carcinoma cell line with a more "consistent history" of BRAF dependence, and identified LS411N as one

such cell line (Jebelli, Baradaran et al., 2021). We find that, similar to DBTRG, knockdown of BRAF and not DUSP4 leads to suppression of cell proliferation (Manuscript Figure 3F). Surprisingly, BRAF knockdown in these cells also did not show a decrease (in fact, an increase is observed) in pERK levels (Supplementary Figure 4D), suggesting that in these cells, negative feedback release may cause pathway reactivation. These new LS411N data have replaced the HT29 data from the original manuscript. As AKT has also been reported to be activated in response to BRAF inhibition in BRAF-mutant colorectal carcinoma cell lines through negative feedback release (Mao et al., 2013) we assessed pAKT levels in LS411N cells following DUSP4 knockdown and found that pAKT levels were indeed increased (Rebuttal Figure 5A). But, in order to demonstrate that MAPK signalling is primarily driven by mutant BRAF in these cells, we took a pharmacological approach and treated cells with increasing doses of dabrafenib with or without trametinib, as this is the current standard of care in mutant BRAF melanomas. We find that 10nM dabrafenib was sufficient to almost completely eliminate ERK phosphorylation (new Supplementary Figure 8A) suggesting that MAPK pathway activation is primarily driven by mutant BRAF in these cells. This is in contrast to the KRAS-mutant/BRAF WT lung cancer cell line PATU8902 or a lung cancer cell line (NCI-H1755) that endogenously expresses a constitutively active but vemurafenib-resistant form of BRAF (BRAF-G469A), both of which show no difference in pERK levels in response to vemurafenib (Rebuttal Figure 5B). We also find that in both LS411N and DBTRG cells, treatment with either dabrafenib alone or dabrafenib in combination with trametinib significantly decreased cell proliferation (new Supplementary Figure 8B) in contrast to PATU8902 and NCI-H1755 which do not show any cell proliferation defects upon Vemurafenib treatment (Rebuttal Figure 5C). These data provides further evidence of MAPK dependence in LS411N and DBTRG cell models.

Rebuttal Figure 5. (A) LS411N cells were transfected with siRNA against non-targeting control, DUSP4 and BRAF. 48h later, cell lysates were analyzed by western blot. (B) NCIH1755 and PATU8902 cells were seeded in the absence of MAPKi and, 24 h later, were treated with Vemurafenib at the indicated concentrations or with DMSO as vehicle. Cell lysates were analyzed by immunoblot after 2h of treatment. (C) NCIH1755 and PATU8902 cells were seeded and 24h later treated with increasing concentrations of Vemurafenib. Cell viability was analyzed by colony formation assay.

Point 4. They state that "the essential role of DUSP4 is restricted to MITF-expressing melanoma and it is independent of the oncodriver mutation". There are NRAS and BRAF melanomas with varying MITF expression independent of the oncodriver mutation. The authors demonstrated that DUSP4 depletion increases pERK either in MITF-high or low melanoma cells, but the effect on cell growth is exclusive to MITF high cells. This suggests that the effects of DUSP4 depletion depend on MITF rather than an optimal level of ERK activity alone. To support their conclusion, the authors used two MITF-high and two MITF-low melanoma cell lines. For consistency, it would be useful to demonstrate that these effects are specific to DUSP4 and not DUSP6 or DUSP10 in SKMEL30 (MITF-high) (S5).

We thank the reviewer for the great suggestion. Following this advice, we have assessed the role of DUSP6 and DUSP10 knockdown in high- and low- MITF expressing cells including SKMEL28, SKMEL30, A375 and SKMEL2 by using siRNAs against DUSP4, DUSP6 and DUSP10 either alone or in combination with siDUSP4 (new Supplementary Figure 6). As discussed in point 1, the efficiency of the DUSP4 and DUSP6 knockdown was evaluated by western blot (new Supplementary Figure 6A). But, due to the low sensitivity of available DUSP10 antibodies, DUSP10 downregulation was assessed by RT-qPCR (new Supplementary Figure 6B). We found that despite further overactivation of ERK following either DUSP4/DUSP6 or DUSP4/DUSP10 double knockdown (compared to single DUSP4 knockdown), cell growth was only impaired in high-MITF expressing cells in response to either dual DUSP4+6/10 or single DUSP4 inactivation (new Supplementary Figure 6C). These results further support the notion that DUSP4 plays a critical role in preventing the superactivation of ERK, which can lead to selective growth inhibition in cells with high MITF levels independently of the oncogenic driver.

Point 5. With respect to the hypothesis that MITF expression levels in treatment-naïve cells determine the essentiality of DUSP4 in MAPKi resistant cells, the authors generated SKMEL28 (MITF-high) and A375 (MITF-low) resistant cells with chronic exposure to lethal doses of Trametinib or Vemurafenib, respectively. In SKMEL28 cells an acute, but not sustained, treatment withdrawal strongly compromised cell viability, but this was not the case for A375 cells with low MITF expression and without variations in DUSP4. It would be clearer if cell viability were shown in all the conditions (Parental, R+T, R-T sustained and R-T acute for SKMEL28; Parental, R+T, and R-T acute for A375) by CellTiter-Glo, as in figure S6A/B.

We thank the reviewer for this great suggestion which we have actioned as follows. SKMEL28 resistant cells were obtained by chronically treating them with lethal increasing concentrations (0.5- 20nM) of Trametinib until resistant pools showed up. During this process, cells developed specific mechanisms of resistance in order to overcome the lethal effect of the drug and be able to survive. Once these cells were able to proliferate normally in the presence of trametinib, they became addicted to it (R+T). Therefore, removal of the drug at short term induced a toxic effect and a massive cell death to SKMEL28 resistant cells (R-T acute). Nevertheless, some cells were able to survive and formed new clones by keeping them longer in culture in the absence of the drug (R-T sustain). In contrast, A375 cells resistant to MAPKi did not show any drug addicted phenotype and, even following drug withdrawal, cells could proliferate normally (R+V). In order to analyze if SKMEL28 and A375 resistant cell lines were resistance to different MAPKi, we treated both cell lines at increasing concentrations of different MAPKi and cell viability was measured by CellTiter-Glo. For this experiment, trametinib and vemurafenib were removed from SKMEL28 R+T and A375 R+V respectively, and cells were seeded in the absence of any drug.

24h later, increasing concentration of different MAPKi were added in all conditions and after three days, cell viability was measured. In the same experiment, parental cells and SKMEL28 R-T sustain were also included. Since cell viability is measured in the presence of all MAPKi overtime, the SKMEL28 R-T acute and A375 R-V acute conditions could not be included in the analysis because they need to be kept acutely in culture without any drug. The rest of the conditions were considered in the experiment (Supplementary Figure 7A, 7B).

The authors state that "these results suggest that the role of DUSP4 is solely critical in high-MITF derived cells with acquired resistance to MAPKi". However, in F5B, they observe a loss of DUSP4 expression and, in consequence, MITF in R+T SKMEL28 cells. In this condition the cells are viable even though there is no DUSP4 to protect them. The authors justify this fact in the discussion suggesting that the MAPKi might be suppressing DUSP4 expression to compensate for the lack of signal. Nevertheless, DUSP4 should be essential for survival, what is the explanation for these cells being perfectly viable without DUSP4, without MITF, and with hyperactivated ERK if the Trametinib dose is a lethal dose instead of the sub-therapeutic dose that would normalize pERK levels?

We thank the reviewer for this comment, and apologise for not having being sufficiently clear on this complex point. Genetic alterations that drive MAPK pathway activation make up about half of the mechanisms of clinical MAPKi resistance. Importantly, this can occur through mutational activation of core pathway components, by upregulation of receptor tyrosine kinases, or through activation of downstream signaling (Luebker & Koepsell, 2019). One of the suggested mechanisms is reduced expression of DUSP4, which has been observed in some resistant cells chronically exposed to MAPKi. Vanishing levels of DUSP4 can reactivate the MAPK pathway, resulting in resistance (Gupta, Bugide et al., 2019, Hugo, Shi et al., 2015). In SKMEL28 R+T cells, DUSP4 expression is indeed significantly suppressed, and P-ERK levels are high despite the presence of trametinib. These cells have become drug addicted and die when the MAPKi is acutely removed due to further overactivation of ERK (Manuscript Figure 5B). However, as the reviewer points out, MITF levels are nearly undetectable in these cells. Notably, it has been reported that a low MITF/AXL ratio predicts early resistance to multiple targeted drugs, and that this drug-resistance phenotype is quite common among mutant BRAF and NRAS melanoma cell lines (Müller, Krijgsman et al., 2014). As shown in Manuscript Figure 5B, AXL (and EGFR) levels are dramatically elevated in SKMEL28 R+T cells, suggesting that these cells have undergone this phenotype switch. Therefore, AXL overexpression together with lack of MITF allows melanoma cells to overcome the lethal effect of MAPKi when cells are under constant drug exposure, likely by expanding the range of MAPK output compatible with viability. Of note, after long-term drug withdrawal in these cells, a surviving population emerges wherein DUSP4 expression and P-ERK levels are restored (SKMEL28 R-T sustain). In this population, cells are once again sensitive to DUSP4 depletion even though they are resistant to MAPKi (Manuscript Figure 5C, 5D) and despite low levels of MITF. This suggests that in the context of acquired MAPKi resistance caused by a transition from a high to a low MITF/AXL ratio, DUSP4 remains a relevant therapeutic target through an MITF-independent mechanism. Consequently, a high MITF level at the time of diagnosis may be a good predictive biomarker of response to DUSP4 blockade throughout the tumour's life cycle. New text has been added on p10 & p14 to discuss these points.

Minor

1. There is a typo in S2E and F figure legend, shDUSP4-1 instead of shDUSP4-2.

We apologise for this error. The typo in S2E and S2F figure legend has been corrected.

2. In the section on possible mechanisms of acquired resistance, a previous study (Reference 35) was cited but the sentence is confusing and might be improved by inserting "it was previously reported".

We apologise for the confusing language. The sentence has been replaced as suggested.

3. The figure numbers are mixed up in the statement on MITF-independent mechanisms (Figure 5B, 5E and 5H). 5H refers to A375 cells.

We apologise for this error. The figure numbers have been corrected in the Results section (Manuscript Figure 5).

REFERENCES

- Ahmed TA, Adamopoulos C, Karoulia Z, Wu X, Sachidanandam R, Aaronson SA, Poulikakos PI (2019) SHP2 Drives Adaptive Resistance to ERK Signaling Inhibition in Molecularly Defined Subsets of ERK-Dependent Tumors. *Cell reports* 26: 65-78.e5
- Alvarez MJ, Shen Y, Giorgi FM, Lachmann A, Ding BB, Ye BH, Califano A (2016) Functional characterization of somatic mutations in cancer using network-based inference of protein activity. *Nat Genet* 48: 838-847
- Cagnol S, Rivard N (2013) Oncogenic KRAS and BRAF activation of the MEK/ERK signaling pathway promotes expression of dual-specificity phosphatase 4 (DUSP4/MKP2) resulting in nuclear ERK1/2 inhibition. *Oncogene* 32: 564-76
- Dilshat R, Fock V, Kenny C, Gerritsen I, Lasseur RMJ, Travnickova J, Eichhoff OM, Cerny P, Möller K, Sigurbjörnsdóttir S, Kirty K, Einarsdóttir BÓ, Cheng PF, Levesque M, Cornell RA, Patton EE, Larue L, de Tayrac M, Magnúsdóttir E, Ögmundsdóttir MH et al. (2021) MITF reprograms the extracellular matrix and focal adhesion in melanoma. *eLife* 10: e63093
- Formstecher E, Ramos JW, Fauquet M, Calderwood DA, Hsieh JC, Canton B, Nguyen XT, Barnier JV, Camonis J, Ginsberg MH, Chneiweiss H (2001) PEA-15 mediates cytoplasmic sequestration of ERK MAP kinase. *Developmental cell* 1: 239-50
- Goding CR (2011) A picture of Mitf in melanoma immortality. *Oncogene* 30: 2304-2306
- Goding CR, Arnheiter H (2019) MITF-the first 25 years. *Genes Dev* 33: 983-1007
- Gupta R, Bugide S, Wang B, Green MR, Johnson DB, Wajapeyee N (2019) Loss of BOP1 confers resistance to BRAF kinase inhibitors in melanoma by activating MAP kinase pathway. *Proceedings of the National Academy of Sciences* 116: 4583-4591
- Huang C-Y, Tan T-H (2012) DUSPs, to MAP kinases and beyond. *Cell Biosci* 2: 24-24
- Hugo W, Shi H, Sun L, Piva M, Song C, Kong X, Moriceau G, Hong A, Dahlman Kimberly B, Johnson Douglas B, Sosman Jeffrey A, Ribas A, Lo Roger S (2015) Non-genomic and Immune Evolution of Melanoma Acquiring MAPKi Resistance. *Cell* 162: 1271-1285
- Jebelli A, Baradaran B, Mosafer J, Baghbanzadeh A, Mokhtarzadeh A, Tayebi L (2021) Recent developments in targeting genes and pathways by RNAi-based approaches in colorectal cancer. *Medicinal research reviews* 41: 395-434
- Jeffrey KL, Camps M, Rommel C, Mackay CR (2007) Targeting dual-specificity phosphatases: manipulating MAP kinase signalling and immune responses. *Nature Reviews Drug Discovery* 6: 391-403
- Karlsson M, Mathers J, Dickinson RJ, Mandl M, Keyse SM (2004) Both nuclear-cytoplasmic shuttling of the dual specificity phosphatase MKP-3 and its ability to anchor MAP kinase in the

cytoplasm are mediated by a conserved nuclear export signal. *The Journal of biological chemistry* 279: 41882-91

Loercher AE, Tank EMH, Delston RB, Harbour JW (2005) MITF links differentiation with cell cycle arrest in melanocytes by transcriptional activation of INK4A. *J Cell Biol* 168: 35-40

Luebker SA, Koepsell SA (2019) Diverse Mechanisms of BRAF Inhibitor Resistance in Melanoma Identified in Clinical and Preclinical Studies. *Frontiers in Oncology* 9

Mao M, Tian F, Mariadason JM, Tsao CC, Lemos R, Jr., Dayyani F, Gopal YN, Jiang ZQ, Wistuba, II, Tang XM, Bornman WG, Bollag G, Mills GB, Powis G, Desai J, Gallick GE, Davies MA, Kopetz S (2013) Resistance to BRAF inhibition in BRAF-mutant colon cancer can be overcome with PI3K inhibition or demethylating agents. *Clinical cancer research : an official journal of the American Association for Cancer Research* 19: 657-67

Müller J, Krijgsman O, Tsoi J, Robert L, Hugo W, Song C, Kong X, Possik PA, Cornelissen-Steijger PDM, Foppen MHG, Kemper K, Goding CR, McDermott U, Blank C, Haanen J, Graeber TG, Ribas A, Lo RS, Peeper DS (2014) Low MITF/AXL ratio predicts early resistance to multiple targeted drugs in melanoma. *Nature Communications* 5: 5712

Papatheodorou I, Fonseca NA, Keays M, Tang YA, Barrera E, Bazant W, Burke M, Füllgrabe A, Fuentes AM-P, George N, Huerta L, Koskinen S, Mohammed S, Geniza M, Preece J, Jaiswal P, Jarnuczak AF, Huber W, Stegle O, Vizcaino JA et al. (2018) Expression Atlas: gene and protein expression across multiple studies and organisms. *Nucleic Acids Res* 46: D246-D251

Seberg HE, Van Otterloo E, Cornell RA (2017) Beyond MITF: Multiple transcription factors directly regulate the cellular phenotype in melanocytes and melanoma. *Pigment cell & melanoma research* 30: 454-466

Smith MP, Rana S, Ferguson J, Rowling EJ, Flaherty KT, Wargo JA, Marais R, Wellbrock C (2019) A PAX3/BRN2 rheostat controls the dynamics of BRAF mediated MITF regulation in MITF^{high}/AXL^{low} melanoma. *Pigment Cell & Melanoma Research* 32: 280-291

Su M, Zhan L, Zhang Y, Zhang J (2021) Yes-activated protein promotes primary resistance of BRAF V600E mutant metastatic colorectal cancer cells to mitogen-activated protein kinase pathway inhibitors. *Journal of gastrointestinal oncology* 12: 953-963

Tanoue T, Moriguchi T, Nishida E (1999) Molecular cloning and characterization of a novel dual specificity phosphatase, MKP-5. *The Journal of biological chemistry* 274: 19949-56

Wellbrock C, Arozarena I (2015) Microphthalmia-associated transcription factor in melanoma development and MAP-kinase pathway targeted therapy. *Pigment Cell & Melanoma Research* 28: 390-406

April 19, 2022

RE: Life Science Alliance Manuscript #LSA-2021-01235-TR

Dr. Igor Vivanco
King's College London
Institute of Pharmaceutical Sciences
150 Stamford Street
Franklin-Wilkins Building (5.81)
London SE1 9NH
United Kingdom

Dear Dr. Vivanco,

Thank you for submitting your revised manuscript entitled "DUSP4 protects BRAF and NRAS-mutant melanoma from oncogene overdose through modulation of MITF". We would be happy to publish your paper in Life Science Alliance pending final revisions necessary to meet our formatting guidelines.

- as pointed out by Reviewer 2, we encourage you to discuss the link between DUSP4 and MITF since it is not yet clear and address the minor comment. No additional experimental work is needed at this point
- please address the remaining Reviewer 3 points
- please provide your manuscript file in editable doc format
- please upload your figures and supplementary figures as single files
- please add an ORCID ID for all corresponding authors; you should have received instructions on how to do so
- please add the Twitter handle of your host institute/organization as well as your own or/and one of the authors in our system
- please consult our manuscript preparation guidelines <https://www.life-science-alliance.org/manuscript-prep> and make sure your manuscript sections are in the correct order; please add a figure legend section after the references
- please use the [10 author names, et al.] format in your references (i.e. limit the author names to the first 10)
- please check your figure callouts; you have a callout for Figure 4F, but this is not in the figure or in legend for figure 4
- please provide source data for figure 4A and S5A

A. FINAL FILES:

B. MANUSCRIPT ORGANIZATION AND FORMATTING:

Sincerely,

Reviewer #1 (Comments to the Authors (Required)):

The manuscript has been successfully revised. All my suggestions/questions have been fully addressed.

Reviewer #2 (Comments to the Authors (Required)):

The authors have addressed most of my comments. However, since they have not been able to rescue the DUSP4 knockdown cells with MITF, the question still remains whether most of the effects observed are simply due to loss of MITF. I would suggest that they try an inducible approach for the rescue study. The link between DUSP4 and MITF is still not clear and should be discussed more clearly.

Minor comment: The authors overuse the terms up/downregulated. It may sometimes be appropriate but is probably not the correct term when discussing "upregulation of phospho-ERK" on page 5 since this is unlikely to be due to the lack of phosphatase activity and thus accumulation of the phospho-form but not active upregulation as the term suggests.

Reviewer #3 (Comments to the Authors (Required)):

The authors corrected the minor revisions and have addressed all the suggestions either giving a fair explanation or further supporting their previous experiments by generating new data. Moreover, they have also mentioned additional bibliography sustaining their statements.

Point 1:

Even though technical replicates are not as significant as biological replicates, it is acceptable to use only one cell line since the only purpose with this primary screen was to find relevant hits that affect melanoma cell growth and the efficiency of these siRNAs was proved in more than one cell line later.

It is understandable that PEA15 is outside the scope of the manuscript, however, since its depletion has a major effect on cell growth, it is important to point it out and justify why the authors did not study this protein further and focused instead only in DUSP4. The addition of this relevant information in the discussion makes the manuscript more robust.

As an extra remark, it looks like DUSP6 protein levels in SKMEL28 (supplementary 4B and 6A) are affected by DUSP10 siRNA and we can observe that also at mRNA levels in COLO829 apart from SKMEL28 (Supplementary 2B).

Point 2:

The authors clarified point 2 with a western blot showing similar high levels of MITF expression in SKMEL28 and COLO829 cells.

Point 3:

The authors explained the lack of impact in pERK levels by negative feedback relief and addressed the confusion that the results in HT29 BRAF-mutant CRC cells raised by using a different non-melanoma cell line found in the literature (LS411N). They replaced the data from the original manuscript being now more consistent with their statement. Furthermore, they substantiated their statement by showing that Vemurafenib treatment does not affect proliferation in lung cancer cells that are non-BRAF-mutant or BRAF-mutant/Vemurafenib-resistant.

Point 4:

The authors demonstrated the specific relevance of DUSP4 in MITF-high melanoma cell lines with more precise experiments adding the new Supplementary figure 6.

Point 5:

The authors provided a complete rationale for the lack of certain conditions in the panel. On the other hand, they render a substantial explanation to the second observation made in this point.

Overall, the authors have made a good job overcoming some weaknesses in their original work, the manuscript has improved and meets now all the requirements to be published in LSA.

Reviewer #1 (*Comments to the Authors (Required)*):

The manuscript has been successfully revised. All my suggestions/questions have been fully addressed.

Reviewer #2 (*Comments to the Authors (Required)*):

The authors have addressed most of my comments. However, since they have not been able to rescue the DUSP4 knockdown cells with MITF, the question still remains whether most of the effects observed are simply due to loss of MITF. I would suggest that they try an inducible approach for the rescue study. The link between DUSP4 and MITF is still not clear and should be discussed more clearly.

As suggested by the editor, we have added some additional text in the discussion to clarify this point (p12 of the revised manuscript).

Minor comment: The authors overuse the terms up/downregulated. It may sometimes be appropriate but is probably not the correct term when discussing "upregulation of phospho-ERK" on page 5 since this is unlikely to be due to the lack of phosphatase activity and thus accumulation of the phospho-form but not active upregulation as the term suggests.

We apologise for the confusing terminology. We have now replaced “upregulation of phospho-ERK” with “accumulation of the phospho-form of ERK” (p5 of the revised manuscript).

Reviewer #3 (*Comments to the Authors (Required)*):

The authors corrected the minor revisions and have addressed all the suggestions either giving a fair explanation or further supporting their previous experiments by generating new data. Moreover, they have also mentioned additional bibliography sustaining their statements.

Point 1:

Even though technical replicates are not as significant as biological replicates, it is acceptable to use only one cell line since the only purpose with this primary screen was to find relevant hits that affect melanoma cell growth and the efficiency of these siRNAs was proved in more than one cell line later.

It is understandable that PEA15 is outside the scope of the manuscript, however, since its depletion has a major effect on cell growth, it is important to point it out and justify why the authors did not study this protein further and focused instead only in DUSP4. The addition of this relevant information in the discussion makes the manuscript more robust.

As an extra remark, it looks like DUSP6 protein levels in SKMEL28 (supplementary 4B and 6A) are affected by DUSP10 siRNA and we can observe that also at mRNA levels in COLO829 apart from SKMEL28 (Supplementary 2B).

We thank the reviewer for this keen observation. DUSP6 protein levels are indeed slightly

downregulated in SKMEL28 upon DUSP10 KD (Supplementary 4B) or in COLO829 at RNA levels (Supplementary 2B). However, these changes do not have any impact on P-ERK levels, and consequently, MITF levels remain unaltered. Additionally, this effect is not observed in other melanoma cells such as SKMEL30, SKMEL2 or A375 (Supplementary Figure 6A), indicating that is more a cell-line-specific effect rather than a general feature of melanoma cells.

Point 2:

The authors clarified point 2 with a western blot showing similar high levels of MITF expression in SKMEL28 and COLO829 cells.

Point 3:

The authors explained the lack of impact in pERK levels by negative feedback relief and addressed the confusion that the results in HT29 BRAF-mutant CRC cells raised by using a different non-melanoma cell line found in the literature (LS411N). They replaced the data from the original manuscript being now more consistent with their statement. Furthermore, they substantiated their statement by showing that Vemurafenib treatment does not affect proliferation in lung cancer cells that are non-BRAF-mutant or BRAF-mutant/Vemurafenib-resistant.

Point 4:

The authors demonstrated the specific relevance of DUSP4 in MITF-high melanoma cell lines with more precise experiments adding the new Supplementary figure 6.

Point 5:

The authors provided a complete rationale for the lack of certain conditions in the panel. On the other hand, they render a substantial explanation to the second observation made in this point.

Overall, the authors have made a good job overcoming some weaknesses in their original work, the manuscript has improved and meets now all the requirements to be published in LSA.

May 5, 2022

RE: Life Science Alliance Manuscript #LSA-2021-01235-TRR

Dr. Igor Vivanco
King's College London
Institute of Pharmaceutical Sciences
150 Stamford Street
Franklin-Wilkins Building (5.81)
London SE1 9NH
United Kingdom

Dear Dr. Vivanco,

Thank you for submitting your Research Article entitled "DUSP4 protects BRAF and NRAS-mutant melanoma from oncogene overdose through modulation of MITF". It is a pleasure to let you know that your manuscript is now accepted for publication in Life Science Alliance. Congratulations on this interesting work.

DISTRIBUTION OF MATERIALS:

Again, congratulations on a very nice paper. I hope you found the review process to be constructive and are pleased with how the manuscript was handled editorially. We look forward to future exciting submissions from your lab.

Sincerely,
